# Experimental warming differentially affects vegetative and reproductive phenology of tundra plants

Courtney G. Collins [1✉], Sarah C. Elmendorf[1], Robert D. Hollister [2], Greg H. R. Henry[3], Karin Clark[4], Anne D. Bjorkman [5], Isla H. Myers-Smith [6], Janet S. Prevéy[7], Isabel W. Ashton[8], Jakob J. Assmann [9], Juha M. Alatalo [10], Michele Carbognani [11], Chelsea Chisholm [12], Elisabeth J. Cooper [13], Chiara Forrester[1], Ingibjörg Svala Jónsdóttir [14,15], Kari Klanderud [16], Christopher W. Kopp[17], Carolyn Livensperger [18], Marguerite Mauritz [19], Jeremy L. May[20], Ulf Molau[5], Steven F. Oberbauer[20], Emily Ogburn[1], Zoe A. Panchen[3], Alessandro Petraglia [11], Eric Post [21], Christian Rixen[22], Heidi Rodenhizer [23], Edward A. G. Schuur [23], Philipp Semenchuk [24], Jane G. Smith[1], Heidi Steltzer[25], Ørjan Totland[26], Marilyn D. Walker[27], Jeffrey M. Welker[28,29] & Katharine N. Suding [1]

Rapid climate warming is altering Arctic and alpine tundra ecosystem structure and function, including shifts in plant phenology. While the advancement of green up and flowering are well-documented, it remains unclear whether all phenophases, particularly those later in the season, will shift in unison or respond divergently to warming. Here, we present the largest synthesis to our knowledge of experimental warming effects on tundra plant phenology from the International Tundra Experiment. We examine the effect of warming on a suite of season-wide plant phenophases. Results challenge the expectation that all phenophases will advance in unison to warming. Instead, we find that experimental warming caused: (1) larger phenological shifts in reproductive versus vegetative phenophases and (2) advanced reproductive phenophases and green up but delayed leaf senescence which translated to a lengthening of the growing season by approximately 3%. Patterns were consistent across sites, plant species and over time. The advancement of reproductive seasons and lengthening of growing seasons may have significant consequences for trophic interactions and ecosystem function across the tundra.

A full list of author affiliations appears at the end of the paper.

High latitudes and elevations are warming much faster than the global average[1,2] with Arctic models suggesting between 3–5 °C of warming in spring and 7–13 °C in autumn by the end of the century[3]. One consequence of this warming is altered plant phenology, including both the initiation and/or duration of vegetative and reproductive phenophases[4–6]. Changes in tundra plant phenology have considerable implications for plant–pollinator interactions, herbivory, productivity, and carbon and energy balances[7]. Despite this, we still lack a broad understanding of how warming influences the timing of multiple plant phenophases, particularly later in the season[8] and whether all phenophases will respond in unison[9], or whether divergent responses to warming across phenophases will change the relative timing and duration of phenological events[5,10]. This is especially true for the tundra, which is among the least studied biomes for plant phenology responses to climate change[11]. Therefore, an improved understanding of how warming influences tundra plant phenology across multiple phenophases is crucial to predicting how tundra ecosystems will function in a warmer world.

There are numerous potential scenarios for how and why plant phenophases may respond distinctly to a warming climate. Reproductive and vegetative phenophases may differ in response to warming temperatures due to differences in plant-level physiology and co-evolutionary drivers (Fig. 1, tissue-type response scenario). For example, leaves and flowers utilize different physiological mechanisms to prevent frost damage and allow for sensitive responses to early spring temperature cues[12,13]. Supercooling is the primary mechanism shown to prevent frost damage in reproductive structures of tundra plants[14,15], while extra-organ freezing or freezing tolerance is most common in leaves[13]. Differing evolutionary mechanisms may also play a role, as flowering has likely co-evolved with pollinators while leaf phenology is influenced by herbivore pressure[16]. In particular, species with early season pollinators (i.e., spring-flowering species) have shown higher phenotypic plasticity and ability to advance flowering with warming than later flowering species[17,18]. Variable temperature sensitivity of leafing and flowering phenophases might also reflect selection for lower temporal overlap with interspecific competitors or herbivores or higher overlap with cross-pollinating conspecifics[19]. Differential responses of flowering and leaf phenology can have consequences across trophic levels[20,21], highlighting the need for experimental comparisons of warming effects on both reproductive and vegetative phenophases.

Early and late season phenophases may also differ in their responses to warming (Fig. 1, early-late response scenario) if they are co-limited by different non-temperature variables such as snowmelt, and/or photoperiod[22–26]. Broadly, long-term remote sensing and in situ monitoring suggest that early season (spring) phenophases will advance with warming[27–29], while general patterns for late season (autumn) phenophases, primarily leaf senescence, remain unresolved[27,30–32]. The lack of resolution in late season phenological responses is due to several factors including conflicting evidence on patterns and drivers of senescence[33–35], and fewer studies overall for autumnal phenophases[8]. Asynchronous shifts in early and late season phenophases may result in the lengthening or contracting of the growing, flowering, and/or fruiting seasons[21,24,36–39], with important implications for primary production and trophic interactions[40,41]. On the other hand, the start and end of plant phenoperiods (growth, flowering, and fruiting periods) may shift in concert because of fixed periodicity in phenophase duration[9,39,42] (Fig. 1, unison response scenario). Thus the net impacts of warming on both the timing and duration of tundra plant phenology require further investigation.

In addition to distinct responses among plant phenophases, overall plant sensitivity to warming can vary across spatial and temporal gradients in climate and resource availability[43–46] which can result in amplified or saturating phenology responses under certain conditions[47–49]. For example, warming can dry out surface soils and increase the rate of snowmelt[50,51], potentially creating abiotic stress and shifting the timing of plant growth and phenology, especially for shallow-rooted or non-vascular plant species[44,52,53]. At colder, high latitude sites, the thermal sums required to trigger phenological events may be lower than at warmer, lower latitude sites and thus phenological responses to the same amount of temperature change may be stronger at higher latitudes[47,49,54] or in colder years at a given site[55,56]. Finally, initial plant responses may differ from long-term responses to warming[43,57], as many tundra plants use stored resources from previous growing seasons to initiate growth or flowering, and these lag effects can delay responses[58].

Experimental warming manipulations, including passive open-top chambers (OTCs) are a widely used method to isolate the influence of warming temperatures on plant phenology[51,59–63]. Plant phenology can be influenced by external cues such as temperature, snowmelt, and day length, and internal physiological cues such as a deterministic leaf or flowering longevity that may be phylogenetically constrained[9,12,64–66]. Experimental approaches are necessary in order to disentangle these multiple, in some cases interacting, drivers and develop clear predictions of plant phenology responses to a warming climate[67,68]. However as with most experimental manipulations, in situ studies of plant phenology are limited in terms of species coverage, spatial extent, time period, and inclusion of multiple phenophases[31,69].

We address these limitations through a synthesis of data from the International Tundra Experiment (ITEX) covering 18 sites (Fig. 2) and 46 OTC warming experiments across Arctic, sub-Arctic, and alpine ecosystems with observations between 1 and 20 years in duration on six plant phenophases (green up, flowering, end of flowering, fruiting, seed dispersal, and leaf senescence). To our knowledge, this is this largest synthesis of experimental warming effects on tundra phenology to date, consolidating knowledge at the biome level, and providing experimental evidence of warming impacts across all major plant phenophases.

We ask the following questions:

1. What is the overall magnitude (number of days) and direction (advance or delay) of plant phenology shifts in response to warming?
2. Does warming differentially affect reproductive and vegetative phenology?
3. Does warming shorten, lengthen, or have no effect on the duration of growth, flowering, and fruiting/seed maturation periods?
4. How do plant responses to warming vary across spatial and temporal gradients in resource availability and ambient climate?
5. Are plant responses to warming sustained over time?

We hypothesize that warming will: (1) Differentially affect reproductive and vegetative phenophases and/or early and late season phenophases (tissue-type, early-late response scenarios, Fig. 1). (2) Shift both the timing and duration of growth, flowering, and fruiting periods. (3) Have stronger effects at higher latitudes and in cold years due to higher plant temperature sensitivity in colder climates. (4) Have stronger effects in dry sites due to interactions between warming and soil water stress and when chambers are deployed year-round due to increased thermal sums and earlier snowmelt. (5) Show enhanced plant responses over time due to initial lag effects of warming on phenology in many tundra species.

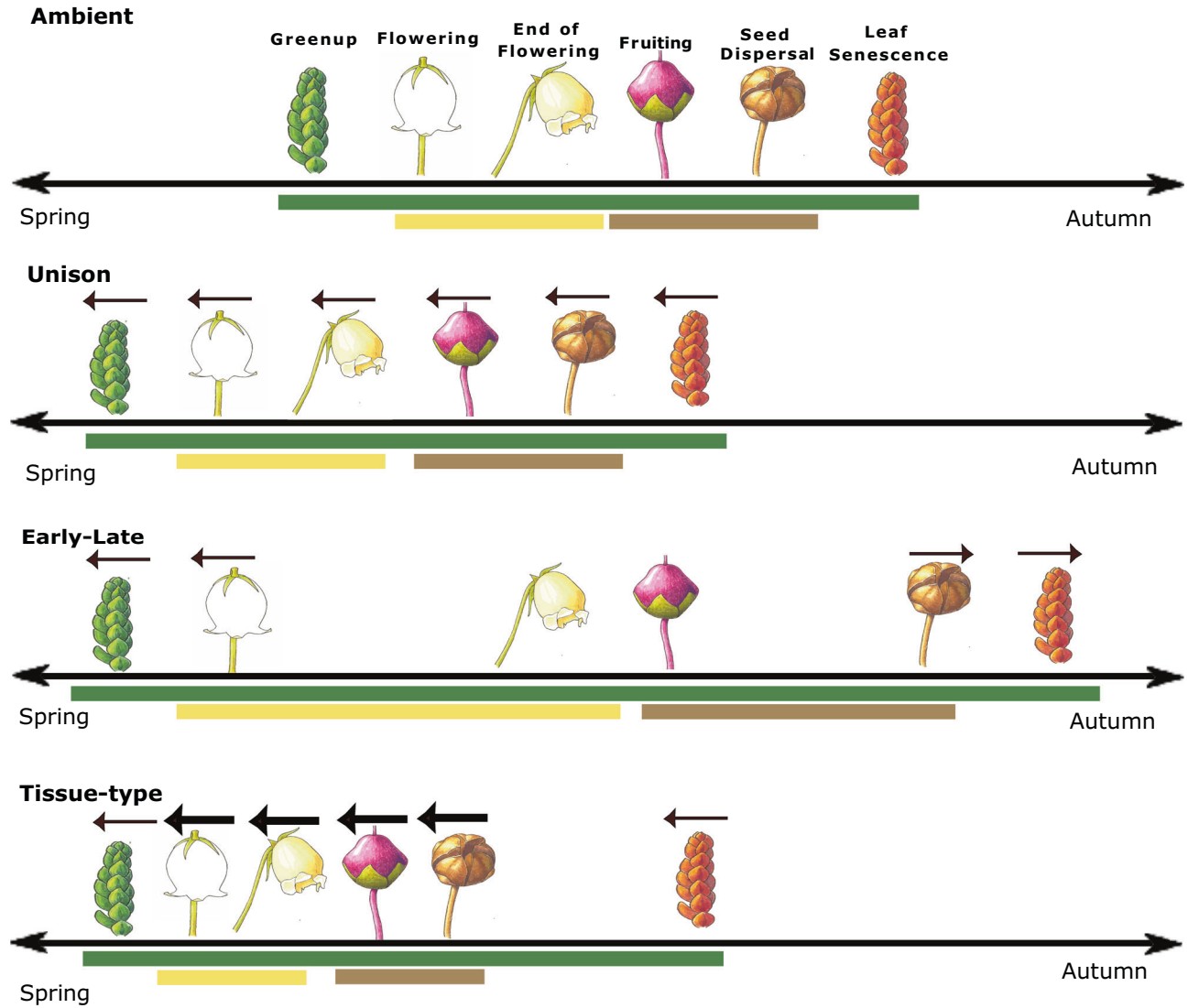

**Fig. 1 Potential scenarios of plant phenology responses to warming.** Phenophases are represented by images, and their timing reflected by their relative position on the line from spring to autumn. The arrows above images show the direction and magnitude (arrow thickness) of changes in timing for each phenophase in response to warming. Green, yellow, and brown horizontal bars reflect the length (duration) of the growth, flowering, and fruiting periods, respectively. Ambient signifies no warming (i.e., control). Unison response scenario: all phenophases shift in the same direction (earlier) by an equal magnitude. The timing of phenology is shifted but there is no change in the duration of phenoperiods. Early-late response scenario: early (spring) and late (autumn) season phenophases shift in different directions (earlier, later) with the same magnitude. The timing of phenology is shifted and the duration of growth, flowering, and fruiting periods are lengthened. Tissue-type response scenario: all phenophases shift in the same direction (earlier) but reproductive phenophases shift by a greater magnitude than vegetative phenophases. The timing of phenology is shifted and the duration of the flowering and fruiting periods are shortened. The first response scenario describes an example of phenology having fixed periodicity where all phenophases shift in concert in response to warming. The second and third scenarios describe examples of distinct responses to warming between early vs. late season or vegetative vs. reproductive phenology, which can result in either a lengthening or shortening of vegetative and/or reproductive periods. This is not an exhaustive list but rather three hypothetical scenarios out of numerous possible plant phenology responses to warming. All botanical illustrations of *Cassiope tetragona* by Jane G. Smith.

Here we show that experimental warming causes: (1) larger phenological shifts in reproductive versus vegetative phenophases and (2) advanced reproductive phenophases and green up but delayed leaf senescence which translates to a lengthening of the growing season by ~3%. Patterns are generally consistent across sites, plant species, and over time, with factors such as ambient climate and soil moisture having minor influences on plant phenological responses to experimental warming. Together, our findings highlight that that climate warming will not simply advance all phases of tundra plant phenology but rather that responses depend on plant tissue type and whether phenophases occur early or late in the growing season. The advancement of species' reproductive seasons and lengthening of species' growing seasons may have potentially significant consequences to trophic interactions and ecosystem function.

## Results

**Experimental warming effects.** We found an effect of experimental warming on five out of six phenophases. Estimated shifts in OTC vs control plots did not overlap zero (based on 90% Bayesian CIs) for green up, flowering, end of flowering, seed dispersal, and leaf senescence (Table 1 and Fig. 3). Four of these phenophases (flowering, end of flowering, seed dispersal,

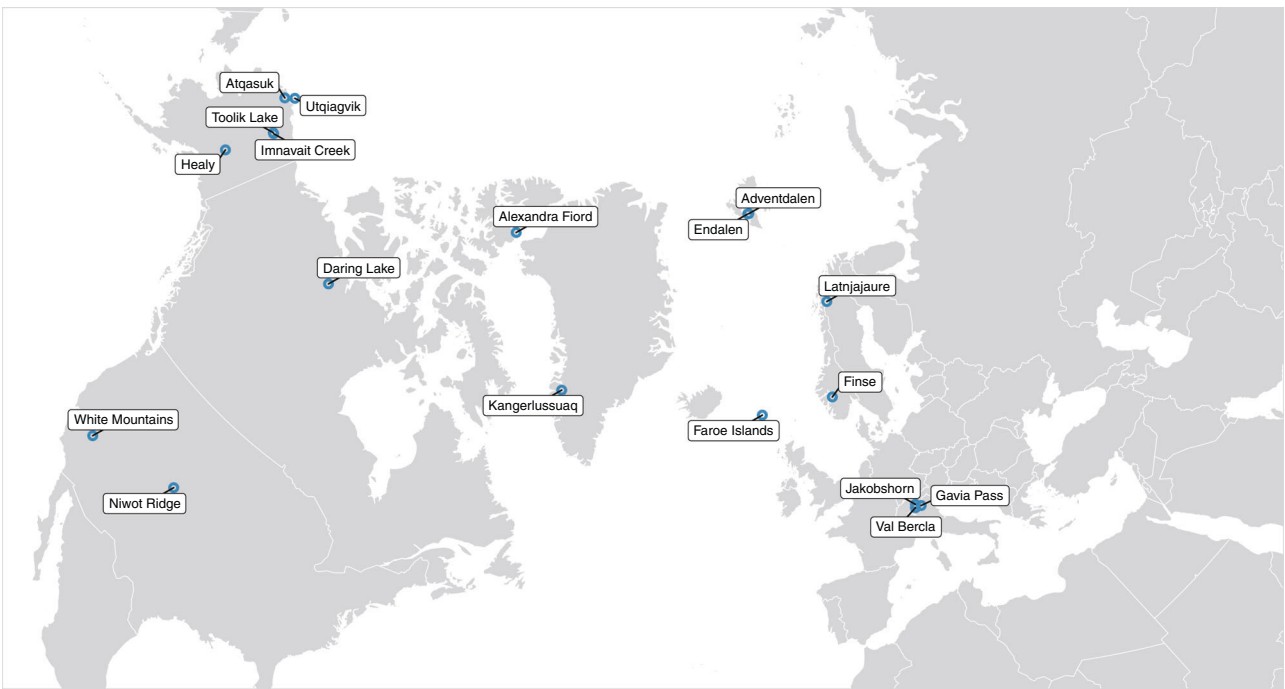

**Fig. 2 Map of study sites.** Map of 18 sampling sites across the ITEX network with warming (OTC) experiments included in this analysis. Created using the sf package (v 0.9.6)[105] in R.

**Table 1 Bayesian hierarchical modeling estimates and credible intervals (90, 95% eti low, high) for the effects of OTC warming on 6 plant phenophases.**

| Model parameter | Phenophase | Estimate (days) | Std. error | 90% eti low | 90% eti high | 95% eti low | 95% eti high |
|---|---|---|---|---|---|---|---|
| OTC | Green up | −0.731 | 0.480 | −1.558 | −0.033 | −1.789 | 0.108 |
| OTC | Flowering | −2.437 | 0.612 | −3.477 | −1.503 | −3.724 | −1.330 |
| OTC | End of flowering | −1.877 | 0.569 | −2.788 | −0.992 | −3.031 | −0.760 |
| OTC | Fruiting | −2.581 | 1.875 | −5.600 | 0.100 | −6.572 | 0.947 |
| OTC | Seed dispersal | −2.902 | 1.399 | −5.187 | −0.702 | −5.737 | −0.179 |
| OTC | Leaf senescence | 0.766 | 0.359 | 0.174 | 1.340 | 0.032 | 1.452 |
| OTC × Soil moisture | Flowering | 1.313 | 0.657 | 0.231 | 2.294 | 0.044 | 2.495 |
| OTC × OTC period | Flowering | 2.191 | 1.075 | 0.506 | 3.975 | 0.171 | 4.406 |
| OTC × Site T | Seed dispersal | −0.880 | 0.525 | −1.790 | −0.063 | −1.984 | 0.065 |

Model parameter signifies the effect of treatment (OTC = difference in days between OTC-CTL) and the interaction of treatment (OTC x) with a spatiotemporal (st) predictor. The interaction of OTC warming with other factors was examined for years of warming, latitude, soil moisture, OTC deployment period, and site mean T and site-year T anomaly; only those interactions where estimates did not cross zero based on 90% Bayesian credible intervals are shown (complete model results can be found in Supplementary Tables 3–6). For the soil moisture interaction, the model parameter signifies the difference in days between moist and dry sites, respectively, showing flowering occurred later in OTCs at moist sites. For the OTC period interaction, the model parameter signifies a difference in days between year-round and summer only OTCs, respectively, and sites where OTCs that were deployed in the summer only had later flowering. For the Site T interaction, the model parameter signifies the difference in days between OTC and CTL plots per degree (°C) in Site T based on species' climate windows, with earlier seed dispersal in OTC plots for species whose dispersal periods coincide with warmer ambient site temperatures.

and leaf senescence) also did not overlap zero at the 95% Bayesian CIs (Table 1). All reproductive phenophases shifted earlier, with 2.4 ± 0.6 days advancement for flowering, 1.9 ± 0.6 days advancement for end of flowering, and 2.9 ± 1.4 days advancement for seed dispersal. Vegetative phenophases shifted both earlier and later, but with a lesser magnitude than reproductive phases supporting a tissue-type response scenario (Fig. 1), with 0.7 ± 0.5 day advancement for green up and a 0.8 ± 0.4 day delay in leaf senescence (Table 1). There was more variation in species' responses for reproductive than vegetative phenophases, as evidenced by wider posterior estimates and confidence intervals for seed dispersal, flowering, and end of flowering versus green up and leaf senescence (Table 1 and Fig. 3). Fruiting showed no consistent response to warming and varied strongly by site and overall (Fig. 3 and Supplementary Table 4).

Estimated shifts of green up and leaf senescence in response to OTC warming were in opposing directions supporting an early-late response scenario (Fig. 1). These differences reflect a lengthening of species' growing seasons by 1.5 ± 0.6 days (Bayesian CIs 90% [0.58, 2.52], 95% [0.38, 2.75]) (Supplementary Fig. 1). Flowering also shifted slightly more than the end of flowering, however, there was no change in the length of species' flowering periods based on Bayesian CIs (90% [−0.78, 1.94], 95% [−1.04, 2.24]). Posterior distributions for fruiting and seed dispersal overlapped entirely suggesting no change in the length of species' fruiting periods (Fig. 3).

**Interactions with spatiotemporal factors.** While warming effects varied substantially among phenophases, within a phenophase, responses were fairly stable across latitudes, moisture regimes, site climate windows, and over time. Spatiotemporal covariates

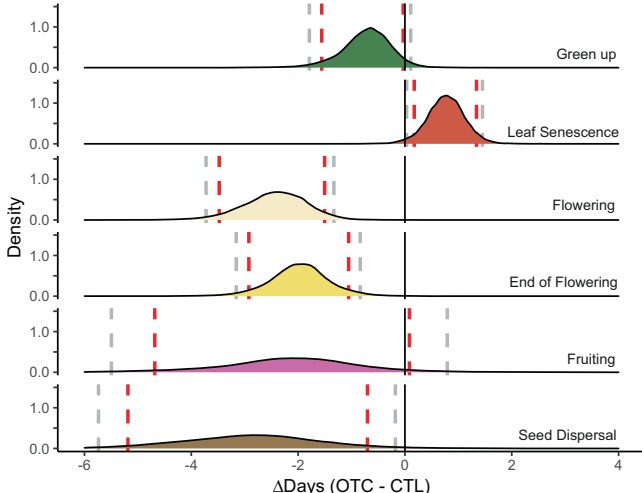

**Fig. 3 Effects of experimental warming on phenology.** Reproductive phenophases shifted with a greater magnitude than vegetative phenophases to experimental warming. Density plots of modeled estimates of treatment effects for the difference (in days) in phenophase timing for plants growing in OTC versus control (CTL) plots. Colors designate each of the 6 measured phenophases. Estimates are shown on the x axis and phenophases are plotted vertically including data from all sites, years, and species as modeled using Eq. 2. Black vertical lines denote zero difference (no change) in the timing of phenology between OTC and CTL plots while red and gray dashed lines denote the 90% and 95% Bayesian credible intervals, respectively. Peaks to the left of black lines indicate an advancement, while peaks to the right of black lines indicate a delay, of that phenophase in response to warming. Created using the ggplot2 package (v 3.3.2)[106] in R.

showed effects for only three out of 42 interactions tested (Table 1 and Supplementary Table 3). Specifically, flowering in OTCs at dry sites was 1.3 ± 0.7 days earlier than in OTCs at moist sites and 2.2 ± 1.1 days earlier in year-round OTC sites versus summer only OTC sites (Fig. 4 and Table 1). Also, seed dispersal was 0.88 ± 0.53 days earlier per degree C ambient air temperature during a species' dispersal period (site $T_{mean}$ species' climate window) (Supplementary Fig. 2).

There was little variation in response to OTC warming at the species, year within site, and subsite within site levels (sd (treatmentOTC), Supplementary Table 4). Variation at the site level was slightly higher, particularly for flowering, fruiting, and seed dispersal (sd(treatmentOTC), Supplementary Table 4). Group level estimates of the effect of treatment (OTC - control) for each species, site, site:subsite, and site:year for all phenophases can be found in Supplementary Table 6a–d.

## Discussion
Our findings demonstrate measurable shifts in the timing of plant phenology for green up, flowering, end of flowering, seed dispersal, and leaf senescence in response to experimental warming across the tundra biome (Fig. 3 and Table 1). Specifically, we observed advances in leaf green up and reproductive phenology of 0.7–2.9 days and delays in leaf senescence of 0.8 days in response to an average of 1.4 °C of warming (0.5–2.3 °C). Furthermore, we find evidence of both Tissue-type and Early-late response scenarios to warming (Fig. 1). Broadly, reproductive phenophases shifted earlier and by a larger magnitude than vegetative phenophases with experimental warming (Fig. 3), and divergent responses of green up and leaf senescence suggest a lengthening of species' vegetative growing seasons (Fig. 3 and Supplementary

Fig. 1). Observed patterns were relatively consistent across sites, plant species, and over time, and spatiotemporal factors such as ambient climate and soil moisture had fairly minor influences on overall responses to experimental warming (Fig. 4, Table 1 and Supplementary Fig. 2, Supplementary Table 4). Taken together these patterns suggest that climate warming will likely advance most tundra plant phenology, but delay senescence, and shorten the time period between leaf emergence and flowering. Deciphering the effects of warming temperatures on plant phenology using only observational data can be challenging due to inherent correlations in environmental drivers[67], yet very few studies have assessed phenology responses to experimental warming at this spatial and temporal scale (but see ref. [70]). In addition, our estimates are likely very conservative, as actual levels of climate warming in the Arctic are predicted to reach 3–5× the magnitude of warming achieved in OTCs[3].

Reproductive phenophases shifted by a greater magnitude than vegetative phenophases in response to experimental warming (Fig. 3 and Table 1) suggesting that tundra phenology responses to warming may be specific to plant tissue-type (Fig. 1, Tissue-type response scenario). Flowering, end of flowering, and seed dispersal had the strongest responses to warming, while phenological shifts in green up and leaf senescence were only ~1/3 the magnitude of those in reproductive phenophases. However, deciphering the mechanisms driving these differences is difficult and requires physiological approaches which are beyond the scope of this study. Previous work addressing these questions has generated mixed results. There is some evidence that reproductive tissues may be less vulnerable to spring frost than vegetative tissues[71] and may thus respond more strongly to changing climate due to lower risk of adverse side effects. However, other work suggests that flowers are more sensitive than leaves to spring frost, likely due to the longer life span of leaves versus flowers in perennials which requires more structural investment in vegetative tissues[72]. Alternatively, co-evolution with pollinators may influence the higher flowering sensitivity in response to warming, as earlier flowering species may receive higher pollinator visitations, but also risk higher chances of frost damage[73,74]. Differential vegetative and reproductive responses to climate are poorly understood and require additional study to link processes at the individual plant scale to ecosystem-level patterns in plant phenology.

Differences in reproductive and vegetative responses to warming may have implications for plant–pollinator, herbivore, and multi-trophic interactions. Despite the relatively small magnitude shifts observed in our study, plant and pollinator interaction networks can shift significantly on very short term (day to day) scales[75,76], and influence the stability and feasibility of ecological communities particularly for ecosystems with very short growing seasons such as the Arctic tundra[77,78] Specialist herbivores which consume strictly vegetative or reproductive plant structures may also have reduced forage availability, or depressed nutrient profiles, particularly early or late in the season[79,80]. Furthermore, if the time between green up and flowering is shortened, plants may have less time to develop resources through leaf photosynthesis (i.e., nonstructural carbon) to support reproductive structures and thus reproductive efforts could be impacted[81]. Thus, a 1.9–2.4 day shift in flowering time may be especially relevant for tundra species with rapid flower development, as the average amount of time between start and end of flowering across all sites was only 18 (±8) days (Supplementary Table 5). However many tundra species utilize previous season's reserves and while the majority of species in our data set flower after leaf emergence, some species flower either at the same time or before leaf green up, in which case this shortening may have a lesser impact.

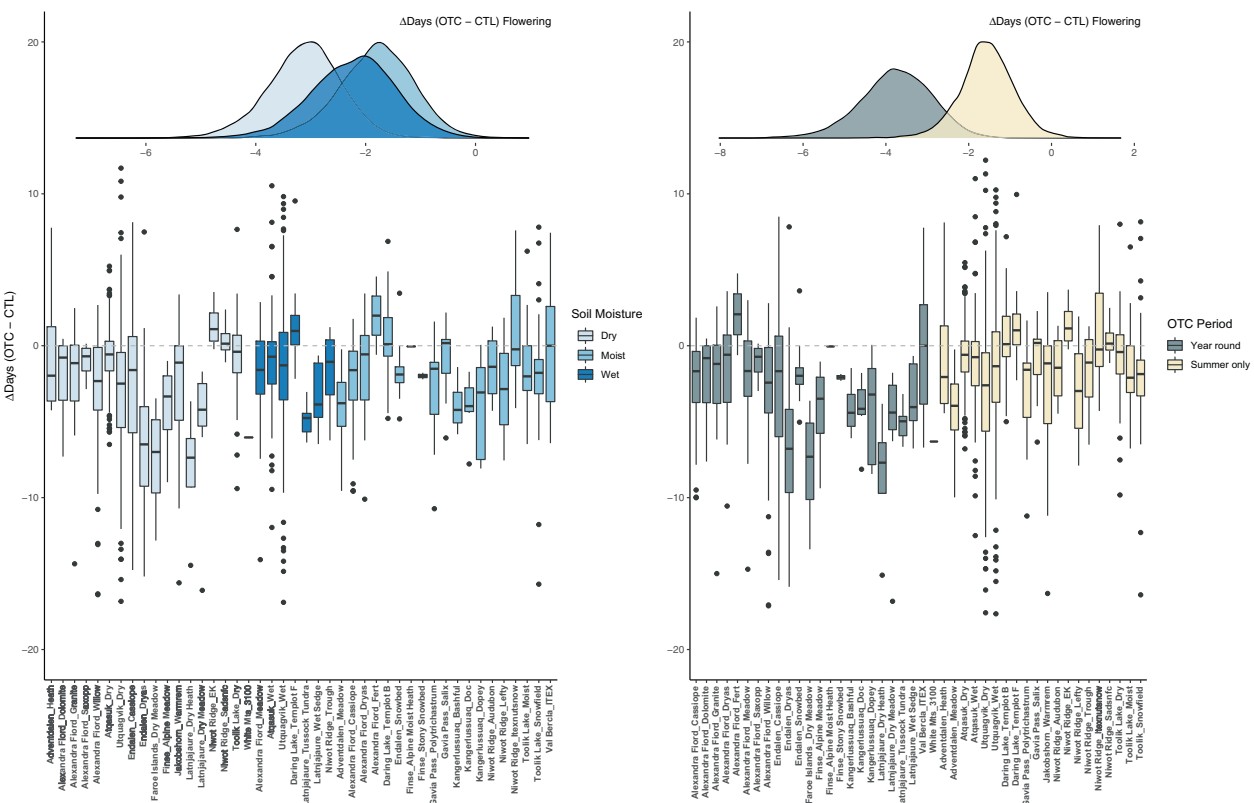

**Fig. 4 Interactions with spatiotemporal factors.** There was a stronger response of flowering to experimental warming in dry versus moist sites and in sites with year-round OTCs versus summer only. Box plots of raw data indicate median (middle line), 25th, 75th percentile (box), and 5th and 95th percentile (whiskers) as well as outliers (single points). Biological replicates (*n*) for each subsite can be found in Supplementary Table 6c. The *y* axis of box and whisker plots and the *x* axis of density plots show the difference (in days) in the timing of flowering for plants growing in OTC versus CTL plots. Box and whiskers plots are plotted by subsite (all species and years). Colors represent either the soil moisture (wet, moist, dry), or the period of OTC deployment (year-round, summer only) for each subsite. Gray dotted lines denote zero difference (no change) in the timing of phenology between OTC and CTL plots, while points above or below these lines denote a delay, or advancement, respectively. Corresponding density plots of modeled estimates for interactive effects of treatment on the timing of plant phenology are shown above box and whisker plots. Created using the ggplot2 package (v 3.3.2)[106] in R.

We also observed divergent responses of green up and leaf senescence to experimental warming (Fig. 3 and Table 1). This partially supports the idea that early and late season phenophases respond differently to warming (Fig. 1, early-late response scenario) with the exception of seed dispersal, which also advanced. Despite far less consensus on autumn phenophases, a meta-analysis of remote sensing and in situ studies suggested a weak delay in fall senescence with climate warming at the global scale, contrasted with strong advancement in spring green up[31]. Our results differ from these patterns as we find a relatively small, but similar magnitude (0.7 and 0.8 days), shift for both green up and leaf senescence with experimental warming, leading to a lengthening at both the beginning and end of species' growing seasons. Inconsistencies between previous work and this study may be due to differences in temperate/boreal and arctic biomes and the fact that observational studies do not isolate the influence of temperature alone[67] In line with our findings, a recent review of Arctic studies showed that green up and flowering tend to advance while leaf senescence is either unchanged or delayed in response to experimental warming[82]. We provide further evidence that leaf senescence is delayed in response to warming in the tundra, helping to resolve this undecided pattern[83]. It is important to note, however, that the shifts we observed were at the average species level, which can differ from the community and ecosystem-level responses, as only a subset of species in the community were sampled at each site. Variation in species'

responses to warming can enhance the likelihood of observing an increase in growing season length at the community or ecosystem level and does not necessarily mean that species' annual life cycles are being extended[38].

Divergent responses of green up and leaf senescence lead to a lengthening species' (vegetative) growing seasons in response to warming. However, we found similar magnitude shifts in the start and end of reproductive phenophases, suggesting no change in the duration of species' flowering or fruiting periods (Fig. 3 and Table 1). Our estimates predict an increase in species' growing season lengths by approximately 1.5 days (Fig. 3 and Supplementary Fig. 1), which reflects a 2.5–3.75% increase in the 40–60 (50 ± 10) day average length between leaf green up and senescence across all sites in this study (Supplementary Table 5). If generalizable across the tundra biome, this level of change could have nontrivial impacts on global carbon (C) stocks, as growing season length is highly correlated with GPP in the Tundra biome[84]. Indeed, modeling studies have shown that a 1-day increase in growing season length can lead to significant increases in annual GPP at mid to high latitudes in the northern hemisphere[41,85]. However, other work has highlighted that gains in production due to an extended growing season do not necessarily translate into enhanced C storage, as autumnal and winter warming are also associated with higher $CO_2$ fluxes from plant and soil respiration over northern latitudes[86,87]. Finally, differential allocation of biomass and phenology above versus below

ground would need to be considered to accurately assess trajectories of carbon storage in tundra ecosystems[88,89].

Plant responses to experimental warming were mostly consistent across gradients of resource availability, climate, and time as we only found three interactions between spatiotemporal factors and experimental warming (Table 1). This highlights that experimental warming can be considered a robust approach to test phenology responses to climate change across multiple species, latitudes, and microclimates. However, we did find interactive effects between experimental warming and soil moisture, OTC deployment period, and site-level climate for flowering and seed dispersal (Fig. 4 and Supplementary Fig. 2). Specifically, flowering responses to experimental warming were stronger in dry sites than in moist sites (Fig. 4), suggesting that warming effects may be enhanced when soil moisture is low and plants are experiencing water stress. This is similar to our finding that flowering was further advanced in sites where OTCs were left on year-round (Fig. 4) as OTCs can cause earlier snowmelt, increased thermal sums, and drier soils than those where OTCs are deployed during the summer only[90,91]. Indeed, in sites where chambers are deployed year-round, snowmelt was advanced by an average of 1.02 days in OTCs versus with control plots (Supplementary Table 7), suggesting that experimental warming may also influence phenology through its effects on snowmelt. However, we only had snowmelt information for a subset of the sites in this study (Supplementary Table 2), and therefore more information is needed to fully assess the interaction(s) between experimental warming and snowmelt on tundra phenology. Overall, our results suggest that warming effects on phenology may be enhanced at dry versus moist sites, and for year-round versus summer warming, with important implications for a predictably drier and more variable tundra biome[92].

We found no evidence that latitude or inter-annual climate variability influenced phenology responses to experimental warming (Supplementary Table 3). However, seed dispersal responses varied by site-level climate, whereby species with warmer climate windows (ambient temperatures during their seed dispersal period) at a particular site, had enhanced responses to experimental warming (Supplementary Fig. 2). In other words, seed dispersal was advanced in warmed compared to control plots overall, and this was further enhanced for species whose dispersal periods coincide with the warmest times of the year. This is the opposite of our initial prediction that plant responses to warming would be weaker in warmer climates[49] or warmer years[55]. However, the two previous studies found decreased effects of warming in warmer climates/years for early season phenophases (green up and flowering) only, while the pattern we observed was for late-season phenophases (seed dispersal) only. This suggests that interactive effects of ambient and experimental warming may differ for early vs. late season phenophases[56], which is intuitive as OTCs will have accumulated higher thermal sums towards the middle and end of the growing season[51].

Finally, phenology shifts observed in our study were consistent over time. We found that plant response to experimental warming did not increase or decline over the years, despite many important processes that can shift species' responses on longer timescales, such as lag effects, local adaptation, thermal acclimation, and changes in resource availability[58,93]. Barrett and Hollister[55] similarly found that overall responses of tundra plants to experimental warming were consistent for 17–19 years at two sites in Northern Alaska. On the other hand, a meta-analysis of reported phenological rates of change in ambient conditions showed that shorter studies tended to produce the strongest estimates of phenological change and vice-versa[47], however, we do not find this to be the case for experimental warming studies.

In conclusion, this study reflects the largest synthesis of experimental warming effects on tundra plant phenology of

which we are aware, consolidating knowledge across this critical yet understudied biome, and generating an improved understanding of the influence of warming temperatures independent from other environmental drivers. We demonstrate that experimental warming creates measurable and consistent impacts on tundra plant phenology, including shifts between 0.7 and 2.9 days earlier and 0.8 days later for the 0.5–2.3 °C of warming achieved in OTC experiments. In addition, divergent shifts in leaf green up and senescence led to a 1.5-day increase in species' growing season lengths, or ~3% of the average 50 days growing season length, given this modest level of warming. We consider these estimates to be on the low end of potential shifts in phenology, as much higher levels of warming (~3–13 °C) are expected in the Arctic by the end of the century[3]. Our work incorporates all major plant phenophases across the growing season, including the beginning and ending phases of leaf, flowering, and fruiting phenology, uniquely testing the impact of warming on both the timing and duration of tundra plant phenology. Our results suggest a lengthening of species' growing seasons, and stronger shifts in reproductive versus vegetative phenophases overall. These patterns support the hypotheses that responses of tundra plant phenology to warming are related to both plant tissue-type and whether an event is early or late in the growing season. We found interactive effects of soil moisture and ambient climate for flowering and seed dispersal, but plant responses to experimental warming did not vary meaningfully across latitude, inter-annual climate, or over time. We provide robust estimates of plant responses to warming across 18 sites, over 100 species, 6 phenophases, and observations between 1 and 20 years in duration. Thus, our study incorporates very large spatial and temporal scales, both of which are crucial in order to estimate accurate effect sizes of global change drivers, and to scale up to predict net ecosystem-level responses[93]. Incorporating these improved phenology estimates into process-based models will generate more accurate predictions of changes in global carbon budgets, albedo, and ecosystem processes in response to climate warming.

We suggest several ways to improve future research and decipher the potential drivers and ecosystem-level consequences of phenology responses to warming across the tundra: (1) Directly testing the physiological cues of vegetative versus reproductive phenology through growth chamber or field manipulations (warming leaves/flowers only), (2) co-measuring individual plant phenology alongside timing of pollinators and assessing outcomes on plant reproductive fitness (seed production and viability), (3) monitoring phenology at the plant community level, where all members (not just dominant or charismatic species) are recorded. (4) Using a gradient of experimental warming treatments (extreme, moderate, mild) to understand the consequences of more severe warming and limits of linearity of responses to temperature and (5) measuring the relationship between changes in phenology and ecosystem C fluxes using in situ plot-level measurements and biomass estimates and/or site-level Eddy covariance.

## Methods

**Experimental design.** We compiled a data set of phenology observations from long-term open-top chamber (OTC) warming experiments and paired control plots within the International Tundra Experiment (ITEX). These data reflect the most up-to-date records from 18 sites and 46 experimental locations within these sites (i.e., subsites) in this network (Table 2 and Fig. 2), across the Arctic, sub-Arctic, and alpine ecosystems with observations from 1992 to 2019 on up to six plant phenophases (green up, flowering, end of flowering, fruiting, seed dispersal, and leaf senescence; Table 3). OTCs are made of fiberglass in either cone or hexagon shape ~1.5–2 m in diameter, however, materials used and chamber size can vary slightly across sites based on plant species being monitored and landscape characteristics[94]. OTCs in this study increased plot level air temperature between 0.5 and 2.3 °C (Supplementary Table 1). Because sites measured the degrees of warming achieved in OTCs at different time periods we are unable to accurately estimate phenology

**Table 2 18 sites and 46 subsites included in this study and the number of species, years, and phenophases that were included from each site.**

| Site | Subsites | Spp | Years | Latitude | Longitude | Phenophases |
|---|---|---|---|---|---|---|
| Alexandra Fiord, NU, Canada | 8 | 7 | 20 | 78.83 | −75.80 | G, F, EOF, FR, SD, S |
| Endalen, Svalbard | 3 | 6 | 4 | 78.18 | 15.76 | F, EOF, SD |
| Adventdalen, Svalbard | 2 | 8 | 1 | 78.16 | 16.10 | G, F, EOF, SD, S |
| Utqiaġvik, AK, USA | 2 | 40 | 20 | 71.28 | −156.60 | G, F, EOF, FR, SD, S |
| Atqasuk, AK, USA | 2 | 28 | 16 | 70.45 | −157.40 | G, F, EOF, FR, SD, S |
| Toolik Lake, AK, USA | 3 | 19 | 8 | 68.63 | −149.60 | G, F, EOF, SD, S |
| Imnavait Creek, AK, USA | 1 | 7 | 3 | 68.62 | −149.32 | G, S |
| Latnjajaure, Sweden | 4 | 8 | 5 | 68.33 | 18.50 | G, F, EOF, SD, S |
| Kangerlussuaq, Greenland | 3 | 7 | 2 | 67.02 | −50.72 | G, F, FR |
| Daring Lake, NT, Canada | 2 | 3 | 19 | 65.87 | −111.53 | F, EOF, FR, SD |
| Healy, AK, USA | 1 | 5 | 6 | 63.88 | −149.25 | G, S |
| Faroe Islands, Denmark | 1 | 1 | 4 | 62.06 | −6.95 | F, EOF |
| Finse, Norway | 3 | 4 | 3 | 60.61 | 7.50 | F, EOF |
| Jakobshorn, Switzerland | 1 | 22 | 1 | 46.77 | 9.86 | G, F, EOF, S |
| Val Bercla, Switzerland | 1 | 12 | 2 | 46.47 | 9.58 | F, EOF |
| Gavia Pass, Italy | 2 | 3 | 5 | 46.34 | 10.49 | F, FR, SD |
| Niwot Ridge, CO, USA | 6 | 19 | 6 | 40.06 | −105.59 | G, F, EOF, S |
| White Mountains, CA, USA | 1 | 1 | 1 | 37.5 | −118.17 | F |

Sites are ordered by latitude (highest to lowest).
G green up, F start of flowering, EOF end of flowering, FR fruiting, SD seed dispersal, S leaf senescence.

**Table 3 Total number of observations, species, sites, subsites, and years, as well as unique species × subsite × year × treatment combinations (replicates) for each of six plant phenophases.**

| Phenophase | Total observations ($i$) | Spp | Sites | Subsites | Years | Replicates ($r$) | Replicates ($r$) climate models |
|---|---|---|---|---|---|---|---|
| Green up | 30,361 | 71 | 11 | 28 | 27 | 1760 | 1526 |
| Start of flowering | 30,011 | 107 | 16 | 44 | 28 | 2782 | 2400 |
| End of flowering | 22,177 | 80 | 13 | 33 | 28 | 2108 | 1846 |
| Fruiting | 17,274 | 53 | 6 | 18 | 28 | 1470 | 1320 |
| Seed dispersal | 8292 | 48 | 9 | 22 | 28 | 770 | 692 |
| Leaf senescence | 17,077 | 61 | 10 | 25 | 27 | 1414 | 1264 |

Models were run separately for each phenophase and climate models had slightly lower replicate numbers due to limits on infilling of daily climate data (see "Climate data" section).

shifts per degree of warming, however, the range of warming achieved is well within the projected climate warming for tundra ecosystems, providing a realistic, though likely conservative, estimate of future scenarios[3,95]. Individual(s) responsible for data collection at each site also classified each experimental location (subsite) into one of three soil moisture classes: dry, containing roughly <20% gravimetric water content (GWC); moist 20–60% GWC; or wet >60% GWC[46] (Supplementary Table 2). When possible, sites recorded the first date(s) in each growing season where plots were snow-free to provide information on snowmelt timing in and out of warming chambers (Supplementary Table 2). Because snowmelt occurs before OTC deployment at summer-only sites, we only test the effect of OTCs on snowmelt for year-round sites (Supplementary Table 7).

Plant species monitored at each site followed criteria as defined in the ITEX manual[94], including prioritizing one or more of the following circumpolar main target species: *Carex aquatillus, Cassiope tetragona, Dryas integrifolia, Dryas octopetala, Eriophorum vaginatum, Oxyria digyna, Bistorta vivipara, Ranunculus nivalis, Salix arctica, Salix herbacea, Salix polaris, Salix reticulata, Saxifraga oppositifolia, Silene acaulis*. Fifteen of the 18 sites included one or more of these circumpolar species, while three sites included one or more dominant tundra plant species present at their sites but not on this list (Supplementary Methods 2).

**Phenology data**. Phenology measurements were taken using a common protocol outlined in the ITEX manual[94], yet sites included slightly different phenology definitions across subsites and species (e.g., flower open vs. bud break), and we included whichever phenophase definition was most commonly measured at a given site for each species across all years. We then grouped these measurements across all sites and categorized them as one of the six standardized phenophases above. All site and species-specific phenology definitions can be found in Supplementary Methods 2. We assumed that male and female flowering time did not differ for dioecious species and we did not separate deciduous and evergreen species' leaf phenology as evergreen species in the Arctic (mostly heath shrubs) also

undergo leaf color change which can be monitored in the same way as deciduous species[30]. In addition, preliminary analyses showed that evergreen and deciduous species did not differ significantly in their timing of leaf phenology overall or in response to OTC warming (Supplementary Table 3).

Sites years, and species varied in census intervals, which can cause bias in the actual estimates of phenological events. For example, if a phenological event is considered to occur on the date it was recorded, this may induce a late-biased estimate. Taking the midpoint between sampling intervals is one way to account for this uncertainty, however, this underestimates the variance in the data set (reviewed in ref. [23]). Therefore, for every phenology event date recorded (day of year, i.e., DOY), we assigned a prior-visit date, which was the most recent recorded visit in the same plot and year prior to the recorded phenology date of interest. For any given observation, if the prior visit could not be assigned with this method (i.e., for data where the phenological event had already occurred by the first visit date: green up 3885 observations, flowering 469 observations, end of flowering 144 observations, other phenophases <25 observations), we took the minimum prior-visit date for each species across all years and then subtracted 3 weeks from that date as a conservative estimate with a minimum of DOY 100 (green up) and 120 (flowering, end of flowering). In addition, missing data can bias effect sizes and therefore we discarded any species × subsite × year combination (green up and leaf senescence only) where more than 20% of the total observations were missing. We then used interval-censored models (see below) to account for inconsistencies in plot monitoring intervals across sites, where the actual date of the phenological event is estimated to occur between the recorded DOY and the assigned prior visit. All species names were updated and standardized using The Plant List (2013, v 1.1) via the package Taxonstand (v 2.3) in R (v 3.6.1)[96].

**Climate data**. We compiled daily mean air temperatures ($T_{mean}$°C) over all measurement years from local weather stations at or near each site (Supplementary Table 1). We averaged these daily temperatures over phenophase-specific climate

windows for each species at a given site (SiteT$_{mean}$°C). Any missing climate data were infilled from ERA5[97] using the methodology described in Kittel et al.[98]. If model selection failed to pick a best model using this method, we used the long-term (multi-year) method to infill (see ref. [98], Appendix B). We allowed for no more than five days of the climate windows to be infilled. Because we used these conservative infilling cutoffs, some site-years were dropped from the climate data set and climate interaction models had slightly lower replicate counts than all other models (Table 3). Second, we calculated site-year temperature anomalies (Site-year$\Delta$T$_{mean}$°C) as the difference between the SiteT$_{mean}$ over all measurement years and the SiteT$_{mean}$ in each measurement year (both averaged over the phenophase-specific climate windows). Climate windows included the 30 days prior to the average day of year (DOY) that each species' phenophase occurred at a given site across all years. For example, if the average flowering date for species $i$ at site $j$ across all measurement years 1 to $k$ was DOY 180, then the Site $T_{mean}$ would be the average daily air temperature from DOY 150–180 at site $j$ across all measurement years, and the site-year$\Delta$T$_{mean}$ would be the difference between the SiteT$_{mean}$ and the average daily air temperature for the same window (DOY 150–180) in each measurement year at site $j$.

**Statistical modeling**. We used a two-step hierarchical modeling approach to test the effects of OTC warming on plant phenology. First, for each phenophase, we estimated the mean DOY and variation in each treatment (OTC or control) × species × subsite × year combination (hereafter replicate, $r$) using interval-censored regression. Prior to regression, we discarded any spp × subsite × year combinations that did not have at least two observations in both OTC and control treatments and removed outliers where the difference in OTC vs. control was greater than 4 standard deviations from the mean for that phenophase. We also standardized the data for each phenophase by calculating the midpoint between prior visit and DOY values for each observation, and subtracting the mean, and dividing by the standard deviation of the midpoints. Next, for each phenophase, we estimated the mean DOYs ($\mu_r$) and standard errors ($\sigma_{\mu r}$) by fitting an intercept-only model (interval-censored) to the data for each replicate (Supplementary Methods 1, Eq.1). Interval-censored models assume the (unobserved) DOY on which each phenophase occurred for each individual observation is normally distributed around the replicate mean $\mu_r$ with variance $\sigma_r^2$ and is estimated using the observed DOYs, censored by the prior visit (DOY) (see "Phenology data" section) in the survreg function of the survival (v 3.2.7) package in R[99,100].

$$Y_i \sim N(\mu_r, \sigma_r^2) \quad (1)$$

The estimates from these models (the mean phenology DOY $\mu_r$ and its standard error $\sigma_{\mu r}$) at the replicate level for each phenophase were used as the inputs to the second stage of the modeling (see below). For a small proportion of the data, we were unable to estimate these terms using the intercept-only model (Eq. 1), because all observations within a replicate had the same DOY and prior-visit values. If this was the case, we used the average of the midpoints between the DOY and prior visit as the mean ($\mu_r$) and the average standard error at the species x subsite x treatment level across all measurement years as the standard error ($\sigma_{\mu r}$).

Second, we used Bayesian hierarchical modeling with default (non-informative) priors in the R package brms (v 2.14.4)[101] to estimate the effects of OTC warming on plant phenology across sites, subsites, species, and years. We used the following model structure to answer questions about mean treatment effects (Q1–3):

$$\mu_r \sim N\left(\alpha + \alpha_{s[r]} + \alpha_{k[r]} + \alpha_{y[r]} + \alpha_{j[r]} + \left(\beta + \beta_{s[r]} + \beta_{k[r]} + \beta_{y[r]} + \beta_{j[r]}\right)*\text{Trt}_r, \sigma_r + \sigma_{\mu r}\right) \quad (2)$$

where the response is the previously estimated replicate level mean DOY ($\mu_r$) and associated standard errors ($\sigma_{\mu r}$) which are incorporated into a joint response variable using the resp_se function in brms (Eq. 2). Treatment (Trt$_r$) is an indicator variable for replicates measured in OTCs (1) or control (0), with random effects of treatment grouped by species ($s$), site ($k$), year within site ($y$), and subsite within site ($j$). We used default brms (i.e., non-informative) uniform flat priors ($-\infty$ $\infty$) for the global intercept ($\alpha$) and slope ($\beta$), and half student_t priors with 3 degrees of freedom and scale = 10 for the variance components $\sigma_r$ and all group-level variances $\sigma_\alpha$ and $\sigma_\beta$.

For each group (species, site, year within site, and subsite within site), the group level coefficients ($\alpha_{group}, \beta_{group}$), and their correlation [$\alpha_{group}, \beta_{group}$] were modeled using a multivariate normal distribution with means of zero and standard deviation $S_{group}$, which is the variance-covariance matrix of each varying intercept and slope and their correlation $\rho_{group}$. Finally, we modeled the correlation matrix $R_{group}$ for the parameter $\rho_{group}$ using an LKJ-correlation prior distribution with $\zeta$ parameter = 1, which constrains the correlation term uniformly between −1 and 1. See Supplementary Methods 1 for full model parameterization.

For questions about variation in treatment effects over space, time, and ambient climate (Q4 and 5), we used the same model structure but with an additional fixed interaction term between treatment and one of six spatiotemporal (St) predictors of interest: (1) Years of warming (continuous, replicate level-$r$) calculated as a number of years from the start of the experiment at each site/subsite, (2) latitude (continuous, site level-$k$), (3) water availability ((categorical: dry/moist/wet) based on gravimetric water content (GWC), site:subsite level-$j$), (4) OTC deployment period (categorical (year-round/summer only), site level-$k$), (5) site mean temperature (continuous, site level-$k$), and (6) site-year temperature anomaly (continuous, site:year level-$y$) (Eq. 3). We included both climate predictors in the

same model using a group means centering approach (described in van de Pol and Wright[102]) with two interaction terms to distinguish spatial (site- $\mu T$) and temporal (siteyear-$\Delta T$) influences of temperature on OTC warming (see "Climate data" section). See Supplementary Methods 1 for full model parameterization.

$$\mu_r \sim N(\alpha + \alpha_{s[r]} + \alpha_{k[r]} + \alpha_{y[r]} + \alpha_{j[r]} + \beta_1 * \text{Trt}_r + \beta_2 * \text{St}_{r,s,k,y,j} + \beta_3 * \text{Trt}_r \times \text{St}_{r,s,k,y,j}$$
$$+ \left(\beta_{s[r]} + \beta_{k[r]} + \beta_{y[r]} + \beta_{j[r]}\right) * \text{Trt}_r, \sigma_r + \sigma_{\bar{y}r}) \quad (3)$$

Models were run separately for each phenophase (Table 3) and each model was run with two chains of 10,000 iterations (warm-up 5000 iterations, no thinning) Markov Chain Monte Carlo (MCMC) sampling (brms default). We checked for convergence of chains for all parameters both visually with trace plots and with the Gelman–Rubin convergence statistic[103]. Trace plots showed that chains mixed well and converged to stationary distributions for all parameter estimates. Gelman–Rubin convergence statistics for parameter estimates of all models were less than 1.1. We calculated Bayesian credible intervals (CIs 90, 95%) for all fixed model parameters in the R package BayestestR (v 0.90)[104] using the equal tailed interval (eti) method and consider modeled parameter estimates to demonstrate an effect on the response variable when Bayesian CIs did not cross zero. For question 3, to determine whether experimental warming will alter the duration of plant phenoperiods, we calculated the difference between posterior distributions of green up and leaf senescence, flowering and end of flowering, and fruiting and seed dispersal for species' growth, flowering, and fruiting periods respectively and also calculated Bayesian CIs for each of these estimates. Because most fruits in this data set are dehiscent, and fruiting is most often defined as the first seed development (Supplementary Methods 2), we defined the fruiting period as the length of time between fruiting and seed dispersal. It is important to note, however, that not all sites, species, years included each of these paired phenophases (see Supplementary Table 6a–d for replicates per phenophase). Finally, we extracted model intercepts from treatment-only models (Eq. 2) to estimate the average DOY when each phenophase occurred across all sites, species, and years and the average number of days between starting and ending phenophases as a baseline with which to compare treatment responses.

**Disclaimer**. Any use of trade, firm, or product names is for descriptive purposes only and does not imply endorsement by the U.S. Government.

**Reporting summary**. Further information on research design is available in the Nature Research Reporting Summary linked to this article.

## Data availability
All data used in this analysis can be found at https://github.com/cour10eygrace/OTC_synthesis_analyses.git. Figures 3, 4 and Supplementary Figs. 1, 2 of this manuscript have associated raw data. The complete data set is also archived in the Polar data Catalog https://doi.org/10.21963/13215.

## Code availability
All analysis code can be found at https://github.com/cour10eygrace/OTC_synthesis_analyses.git[107].

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

## Acknowledgements

We thank Maja Sundqvist, Aimee Classen, Nathan Sanders, Warren Sconiers, Nyika Campbell, William Bownman, and many additional field assistants for their help in collecting phenology data. We thank Dr. Jessica Savage for helpful comments and feedback on an earlier version of the manuscript. We thank Caitlin White for assistance with the climate infilling code. Funding was provided by the following: Norwegian Research Council ("SnoEco" project, number 230970), the FRAM Centre Terrestrial Flagship ("SnoEcoFen" project), and the Norwegian Centre for International Cooperation in Education (SIU) High North Programme ("JANATEX" project, number HNP2013/10092) to Elisabeth J. Cooper. A.W. Garfield Weston Foundation Postdoctoral fellowship to Zoe A. Panchen. The U.S. Department of Energy, Office of Biological and Environmental Research, Terrestrial Ecosystem Science (TES) Program Award #DE-SC0006982, #DE-SC0014085, #DE-SC0020227, and an NSF PLR Arctic System Science Research #1931333; NSF NNA: LTREB Award # 1754839 to Edward (Ted) Schuur and Marguerite Mauritz. The National Science Foundation Office of Polar Programs Grants #PLR-1007672, 0902096, and 0902184 to Heidi Steltzer. National Science Foundation Graduate Research Fellowships (GRFP) to Carolyn Livensperger and Chiara Forester. A Semper Ardens grant from the Carlsberg Foundation to Chelsea Chisholm. Research Experience for Undergraduates (REU) Program funding to Emily Ogburn. National Science Foundation grant #'s 9907185, 632277, 856710, 1432982, 1504381, and 1836898 to Steven Oberbauer and Jeffrey Welker. National Science Foundation grant #'s 9714103, 632263, 856516, 1432277, 1504224, 1836839 to Robert Hollister. A UK Natural Environment Research Council ShrubTundra Grant (NE/M016323/1) to Isla Myers-Smith. A Stiftelsen Oscar och Lili Lamms Minne research grant to Juha Atalo. Additional funding provided by the Government of the Northwest Territories, the University Centre in Svalbard. Publication of this article was funded by the University of Colorado Boulder Libraries Open Access Fund, Katharine Suding, and Bob Hollister. The appropriate permits to access research sites were obtained whenever necessary and the permits/permissions necessary varied among the different sites. Specific permitting information providing access to field sites is as follows: Adventdalen: Store Norske Spitsbergen Kullkompani A/S 06/792/051.5/PCF and Longyearbyen Local Styre (2009) 401-2 sak 34/09, Alexandra Fjord: Qikiqtani Inuit Association and Nunavut Dept of Environment (1989), Gavia Pass: Stelvio National Park (2008), Latnjajaure: Abisko Scientific Research Station permission for long-term experiment "Linking plant and soil ecology" (1994), Kangerlussaq:Government of Greenland (2002), Niwot Ridge: University of Colorado Mountain Research Station and Arapaho and Roosevelt National Forests Special Use Permit (1994), White Mountains: Inyo National Forest Special Use Permit (2014), Endalen: Longyearbyen Lokal Styre (2014) 456-2-X70, Toolik Lake: Bureau of Land Management Alaska Northern Field Office (1994), Atqasuk & Utqiagvik: Ukpeagvik Iñupiat Corporation (1994), Healy: Site permissions under LAS-24220 issued to the University of Alaska by the State of Alaska Department of Natural Resources (2004), Jakobshorn & Val Bercla: Swiss Federal Institute for Forest, Snow and Landscape Research WSL (1992), Daring Lake: Wek'eezhii Land and Water Board (2009), Imnavait Creek: Bureau of Land Management Central Yukon Field Office (2009)#FF09S602, Faroe Islands: Museum of Natural History of the Faroe Islands (2001), Finse: Hallingskarvet National Park Board (2006).

## Author contributions

Data were collected and provided by J.A., A.B., J.G.S., C.F., I.A., E.O., M.W., K.C., I.S.J., R.H., G.H., Z.P., U.M., E.P., K.K., Ø.T., M.C., A.P., J.M.W., S.F.O., M.M., E.S., H.R., H.S., C.L., E.J.C., P.S., C.C., C.R., C.K., and J.M. Conceptual design of the manuscript was performed by C.G.C, S.C.E., A.B., I.M.-S., J.P., and K.S. Data processing and statistical analyses were performed by C.G.C. and S.C.E. C.G.C. wrote the manuscript and created figures, with help from J.A. and J.G.S. All co-authors provided feedback and comments on the first manuscript version and have approved the revised version.

## Competing interests

The authors declare no competing interests.

## Additional information

[1]Institute of Arctic and Alpine Research, University of Colorado Boulder, Boulder, CO, USA. [2]Department of Biology, Grand Valley State University, Allendale, MI, USA. [3]Department of Geography, University of British Columbia, Vancouver, BC, Canada. [4]Department of Environment and Natural Resources, Government of the Northwest Territories, Yellowknife, NT, Canada. [5]Department of Biological and Environmental Sciences, University of Gothenburg, Gothenburg, Sweden. [6]School of GeoSciences, The University of Edinburgh, Edinburgh, UK. [7]U.S. Geological Survey, Fort Collins, CO, USA. [8]National Park Service, Inventory & Monitoring Division, Rapid City, SD, USA. [9]Department of Biology, Aarhus University, Aarhus, Denmark. [10]Environmental Science Center, Qatar University, Doha, Qatar. [11]Department of Chemistry, Life Sciences and Environmental Sustainability, University of Parma, Parma, Italy. [12]Department of Environmental Systems Science, ETH, Zurich, Switzerland. [13]Department of Arctic and Marine Biology, The Arctic University of Norway UiT, Tromsø, Norway. [14]Department of Life- and Environmental Sciences, University of Iceland, Reykjavík, Iceland. [15]The University Centre in Svalbard (UNIS), Longyearbyen, Svalbard, Norway. [16]Department of Ecology and Natural Resource Management, Norwegian University of Life Sciences, Ås, Norway. [17]Biodiversity Research Center, University of British Columbia, Vancouver, BC, Canada. [18]National Park Service, Capitol Reef National Park, Torrey, UT, USA. [19]Department of Biological Sciences, University of Texas at El Paso, El Paso, TX, USA. [20]Department of Biological Sciences, Florida International University, Miami, FL, USA. [21]Department of Wildlife, Fish, & Conservation Biology, University of California Davis, Davis, CA, USA. [22]Swiss Federal Institute for Forest, Snow and Landscape Research WSL, Davos, Switzerland. [23]Center for Ecosystem Science and Society, Northern Arizona University, Flagstaff, AZ, USA. [24]Department of Botany and Biodiversity Research, The University of Vienna, Vienna, Austria. [25]Department of Environment and Sustainability, Fort Lewis College, Durango, CO, USA. [26]Department of Biological Sciences, The University of Bergen, Bergen, Norway. [27]HOMER Energy by UL, Boulder, CO, USA. [28]Department of Biological Sciences, The University of Alaska Anchorage, Anchorage, AK, USA. [29]Department of Ecology and Genetics, The University of Oulu, Oulu, Finland. ✉email: Courtney.collins@colorado.edu

