## [Peer Review File · Nature Communications]

Reviewers' Comments:

Reviewer #1:

Remarks to the Author:

16 Nov 2020

Experimental warming differentially affects vegetative and reproductive phenology of tundra plants

In this paper, the authors synthesize phenology data for tundra plant species collected within the ITEX experimental network. Six 'phenophases', including reproductive phases (e.g., flowering) and vegetative phases (e.g., leaf senescence) were defined, and it was tested how the timing of these phases differed between OTCs and ambient plots. Then, the timing of these phases was used to calculate the length of the growing season, flowering period, and fruiting period. In framing their questions and hypotheses, a conceptual model was used (Fig. 1) illustrating potential scenarios of synchronized or divergent phenophase responses. It was also tested whether or not the plant phenological responses depended on ambient climatic conditions, soil moisture, and experimental conditions (i.e., years of experimental warming and seasonal vs. continuous OTC deployment). Across all sites and across all species, most phenophases advanced by a couple of days, while leaf senescence was slightly delayed. Together, this resulted in increase of the growing season length by, on average, 1.5 days. The phenological responses were largely consistent across the dataset with some small effects of soil moisture, OTC deployment method, and ambient climate.

I enjoyed reading the manuscript. It is generally well-written (but see some of my comments below), and the findings are interesting for a broad audience of climate change ecologists because of their generality. The take-home-message is simple and straightforward. The generality, however, comes at the cost of a lack of any new mechanistic insights. Also, while the findings certainly are of interest, they are not particularly surprisingly or novel. It should be better explained what new we can learn from this study (e.g., compare with Bjorkman et al. 2020_Ambio).

Further, I have a number of concerns:

1) It is not clear how Hypothesis 2 differs from Hypothesis 1. The 'timing' in Hypothesis 1 is also addressed in Hypothesis 1, right? Here, it would help to more explicitly refer to Fig. 1.

2) The conceptual framework presented in Fig. 1 is helpful in understanding the possible outcomes. However, for the 'Early-Late' scenario, the expectation that fruiting would be advanced while seed dispersal would be delayed does not seem to make sense. Why would that be? Any reason why seed ripening would slow down with increasing temperature? I am missing any justification. If fruiting would be advanced, then I would expect seed dispersal also to be advanced. This is actually also what the data shows (no surprise).

3) Snow. OTCs typically accumulate (a lot of) snow. For those sites where OTCs were deployed year round, this could be an important confounding factor. Snow melt is an important driver of plant phenology in tundra ecosystems. Yes, year-round deployment may lead to increased thermal sums (L138), but snow accumulation could have a much stronger (and potentially opposite) effect. For which of the sites were the OTCs deployed year round, and to what extent did this affect snow melt date? Also, in this context, hypothesis 4 needs to be better introduced.

4) As I understand, all phenological responses presented in this manuscript are based on the average of all selected plant species. However, this has not been clearly explained. Also, from the Methods section it is unclear *how* the plant species were selected and *how many* plant species were selected. It is only in the Discussion that I learn that only a subset of the species in the community was sampled. Much more information is needed.

5) The assumption that evergreen and deciduous species do not differ in their leaf phenology does not make any sense. Evergreen and deciduous species per definition differ in leaf phenology. So, these two groups of species should be considered separately.

6) More generally, I understand that the dataset is complicated and that the analyses depend on a lot of assumptions and uncertainties (e.g., L356-374 and L377-394), but without taking into account important variation among sites, species, etc. the results are simply not very informative. One of the strengths of the ITEX network is the availability of all sorts of ancillary data. These data should be better utilized.

7) What are 'subsites'? The study sites are nicely indicated on the map in Fig. 2, but nowhere in the manuscript I can find anything about 'subsites'. What are they, how are they selected/analyzed, and why is it this important?

8) Why would phenology be less variable in OTCs than in control plots (L173-177)? Would this have anything to do the shape of temperature response curves? Some more explanation here would be helpful.

9) The effect of OTCs on air temperature is highly variable among sites (see Tables S1). So, in explaining the large variation in plant phenological responses to OTC warming, it would make sense to actually use this information. Lumping responses to +2.3C in Finse (Norway) with responses to 0.5C in Daring Lake (Canada) is an over-simplification and masks ecologically-relevant differences among sites. Why would you use 'OTC warming' as a categorical variable (yes, no) if you know that in reality is it a continuous variable?

10) The suggestion that early and late season phenological events may differ in their responses to warming 'if they are co-limited by different non-temperature variables' (L84) is interesting. However, other than grouping the sites in 'dry', 'moist', and 'wet' (using unclear criteria; see comment nr. 11) , this idea remains unexplored. Given the large amount of environmental data collected within the ITEX network, I would expect more in-depth and more direct analyses of how 'non-temperature variables' mediated the OTC effect sizes on plant phenological responses. Here, snow melt would be a good start (see comment nr. 3). This would allow to dig deeper into the mechanisms.

11) 'Soil moisture' was used as one of the explanatory variables (e.g., Fig. 4). But, how was 'soil moisture' measured? And how were sites categorized? This is not a trivial task. Nothing is mentioned about this.

12) For the tested 'Year' effects it is unclear what temporal scale this applies to. While the Abstract says that data for 20+ years were used, nowhere in the manuscript I can find any information on this. For all species, were phenological data collected every year for 20+ years? Probably not. Discussion on local adaptation, acclimation, etc. does not make sense if the temporal scale and resolution is unknown. More details and justification is needed.

13) The conclusions about potential consequences for trophic interactions, C cycling, and ecosystem functioning are rather suggestive and are not based on the data presented in this manuscript. The 'significant consequences' of phenological shifts by a day or two are not well substantiated. Paragraph L252-262 is full of uncertainties, and in the end it seems that nothing can be concluded regarding the impacts on global C stocks. Also, the manuscript does not contain any C data. This is despite the large amount of C data collected within the ITEX experimental network.

14) In the Conclusions section, the extrapolation based on *linear* temperature responses makes little sense. For many reasons, temperature responses will certainly not be linear.

15) In the Conclusions section, rather than just suggesting that deciphering the potential drivers and ecosystem level consequences is important (we've known that for many years), it would be much more interesting to propose a way forward on *how* to do that.

Reviewer #2:

Remarks to the Author:

The authors of this report analyse a large dataset including very large spatial and temporal scales to predict ecosystem responses to warming using open-top chambers (OTCs) across Arctic, sub-Arctic, and alpine ecosystems gathered by the ITEX consortium in 1992-2019.

They show that OTCs influence plant phenology, including phenophase shifts in the range of 0.7-2.7 days in response to the 0.5-2.3°C of warming achieved with chambers. However, the high variability of environments prevented to accurately estimate phenology shifts, as OTCs and sites were quite diverse. In addition, phenology definitions slightly differed across subsites and species, and census intervals also varied, introducing more noise in the data. Handling such heterogeneous dataset is commendable, and despite these difficulties, the authors got the most out of the available data.

However, including all sites in the same analyses may only blur the outcome. I wonder whether the different sites or subsites could be grouped to get more meaningful results. I am sceptical that climate in the Alps or Sierra Nevada in California is comparable to the climate in Svalbard or Alexandra Fiord. In addition, by including a large number of plant species assuming that all behave the same way might be an important confounding factor, as they respond in quite idiosyncratic ways to environmental clues.

I am not familiar with some statistical procedures, particularly Bayesian methods, but analyses seem to be sound. The reported rates of change in phenophases differ between control and OTC treatments, although these differences are small, and much larger than inter-annual variations. Thus, I wonder whether such small changes may be meaningful.

I think the manuscript is well written. The results are properly discussed in the context of previous literature, but I think there is some speculation regarding response mechanisms, as they were not addressed in the experiments and require physiologically-oriented approaches. Therefore, this part of the discussion should be drafted more carefully as there are no data to support claims.

Overall, the net contribution of this paper seems not so novel to me, since similar findings have been reported in the last 10-15 years from a diversity of environments, mostly temperate but also high-latitude environments. None of them had, to my knowledge, the strong experimental support used in this paper. Although the reported results support earlier claims concerning phenology responses to climate change, they would certainly be of interest to others in the field. Even if largely confirmative, the data reported in this paper will consolidate knowledge in the field.

Reviewer #3:

Remarks to the Author:

By using an impressive dataset derived from the ITEX initiative, this study analyze detailed both reproductive and vegetative phenological records taken on tundra plant species exposed to experimental warming in 18 sites and 46 sub-site locations. On each location experimental warming was generated passively by using open-top transparent plexiglass chambers, containing vegetation patches formed by different tundra species. The duration of the warming treatment differed among sites, while in some sites phenological records has been taken for 27 yrs. in some

sites there were only one or 2 yrs. of records. The main question addressed with these data set were: 1. What is the overall magnitude (number of days) and direction (advance or delay) of plant phenology shifts in response to warming? 2. Does warming differentially affect reproductive and vegetative phenology? 3. Does warming shorten, lengthen, or have no effect on the duration of growth, flowering and fruiting/seed maturation periods? 4. How do plant responses to warming vary across spatial and temporal gradients in resource availability and ambient climate? 5. Are plant responses to warming sustained over time?

In general, this is very interesting study, addressing important questions of general interest in the scientific community. Nevertheless, although I am very sympathetic with this study, I think that there are some important points that needs to be addressed.

First, it is not totally clear to me the step forwards in our knowledge of the tundra plant phenological responses to warming considering the previous contributions of Prevey et al. (2019: Warming shortens flowering seasons of tundra plant communities. *Nature Ecology & Evolution*, 3(1), 45-52.) and Myers-Smith et al. (2019: Eighteen years of ecological monitoring reveals multiple lines of evidence for tundra vegetation change. *Ecological Monographs*, 89(2), e01351). It is important that authors can clearly state what is really novel in this study.

Second, according to the information provided on Table 3, none of the sites contain 27 yrs. of records for the different phenophases; only 4 of them have a data set longer than 10 yrs., and 9 longer than 5 yrs.; indeed, 5 sites have only 2 or less yrs. of records. Thus, I'm wondering how this asymmetry in the duration of the phenological records can affect the results on either direction. For instance, what if short-term records buffer the long-term responses generating less dramatic changes; or the other way around: short-term records leveraging the observed changes? I think that is important that authors make clear how do they managed the asymmetry in the duration of the phonological records among locations. This is also important for the assessments related to question 4.

Third, the observed changes in some phenophases were within 1-3 days, which are regarded as considerable by the authors. However, I miss more a thorough discussion considering strong examples that shifts of 1-3 days have important consequences for plants. Arguments like those provided in lines 208-2016 are speculative and do not provide unequivocal examples of how shifts of 1-3 days really affect plant performance.

REVIEWER COMMENTS

Reviewer #1 (Remarks to the Author):

16 Nov 2020

In this paper, the authors synthesize phenology data for tundra plant species collected within the ITEX experimental network. Six 'phenophases', including reproductive phases (e.g., flowering) and vegetative phases (e.g., leaf senescence) were defined, and it was tested how the timing of these phases differed between OTCs and ambient plots. Then, the timing of these phases was used to calculate the length of the growing season, flowering period, and fruiting period. In framing their questions and hypotheses, a conceptual model was used (Fig. 1) illustrating potential scenarios of synchronized or divergent phenophase responses. It was also tested whether or not the plant phenological responses depended on ambient climatic conditions, soil moisture, and experimental conditions (i.e., years of experimental warming and seasonal vs. continuous OTC deployment). Across all sites and across all species, most phenophases advanced by a couple of days, while leaf senescence was slightly delayed. Together, this resulted in increase of the growing season length by, on average, 1.5 days. The phenological responses were largely consistent across the dataset with some small effects of soil moisture, OTC deployment method, and ambient climate.

I enjoyed reading the manuscript. It is generally well-written (but see some of my comments below), and the findings are interesting for a broad audience of climate change ecologists because of their generality. The take-home-message is simple and straightforward. The generality, however, comes at the cost of a lack of any new mechanistic insights. Also, while the findings certainly are of interest, they are not particularly surprisingly or novel. It should be better explained what new we can learn from this study (e.g., compare with Bjorkman et al. 2020_Ambio).

RESPONSE: We thank the reviewer for their careful reading of the manuscript and thorough comments. We have addressed each of the reviewer's comments below (in red) and feel the manuscript is thus improved. All updates to the manuscript and supplement are highlighted in yellow.

As the reviewer mentions, it was indeed our intention for the take home message to be simple and straightforward; nonetheless, we believe this study consolidates knowledge in the field and is thus novel and timely for several reasons:

1. This study focuses on tundra plant phenology and is one of the largest analyses of tundra phenology to date. Tundra phenology has been largely understudied in terms of phenology research, second only to the Desert biome (Diepstraten et al. 2018) and thus calls for larger synthesis of this important biome have recently been made. We feel this work addresses this important research gap, in particular the need for multi-site syntheses.

2. Our study focuses on plant phenology responses to experimental manipulations. The need for experimental (not simply observational) estimates of phenology shifts in order to accurately parameterize process based-phenology and global change models has been highlighted in recent reviews (Hanninen et al 2019). This allows us to generate a mechanistic understanding of the influence of warming temperatures independent from other environmental drivers.
 - a. This is particularly critical for late season phenophases such as leaf senescence and fruit maturation/seed dispersal, for which experimental data are greatly lacking (Gallinat et al 2015, Loe et al 2020). Our study includes the most robust experimental estimates of late season phenophase responses to warming in the Tundra biome to date.
 - b. While conceptually similar, Bjorkman et al 2020 *Ambio* was a review of published studies while this is a quantitative synthesis, generating actual parameter estimates of phenology responses to experimental conditions. In addition, Bjorkman et al 2020 (a) only included 3 of the 6 phenophases in our study (leaf emergence, senescence, and flowering) (b) used only published papers which may show some publication bias and (c) used a vote-counting approach which cannot be translated into the magnitude of effects. Here we have used raw data from both published and unpublished studies avoiding these issues. Nonetheless, for consistency we compare our results to the patterns shown in Bjorkman et al 2020 (L249-252).
3. This study incorporates all major plant phenophases across the growing season, including the beginning and end of leaf, flowering, and fruiting phenology. This is more than any other synthesis we are aware of, as many multi-site syntheses analyze one or two plant phenophases (L117) (often leaf green up and/or flowering). For example, Prevey et al (2019) focused on flowering and end of flowering only.

We have worked to more thoroughly incorporate these key points of novelty and how our study moves beyond previously published work throughout the manuscript in the abstract (L9-11) , introduction (L123-125) and discussion/conclusion (L198-200, 323-326, 335-337) .

Further, I have a number of concerns:

1) It is not clear how Hypothesis 2 differs from Hypothesis 1. The 'timing' in Hypothesis 1 is also addressed in Hypothesis 1, right? Here, it would help to more explicitly refer to Fig. 1.

RESPONSE: Yes, we agree with the reviewer that timing was addressed in both H1 and H2. For clarity, we have combined them and made H1 about changes in timing and H2 about changes in duration. We have also added references to how these hypotheses correspond with the different scenarios in Fig 1 and removed the potential mechanisms driving these hypothesized shifts, as we have not directly tested them in this study. This section now reads (L136-138):

“We hypothesized that warming will: 1) Differentially affect reproductive and vegetative phenophases and/or early and late season phenophases (‘Tissue-type’, ‘Early-late’ response scenarios, Fig 1). 2) Shift both the timing and duration of growth, flowering and fruiting periods. “

2) The conceptual framework presented in Fig. 1 is helpful in understanding the possible outcomes. However, for the ‘Early-Late’ scenario, the expectation that fruiting would be advanced while seed dispersal would be delayed does not seem to make sense. Why would that be? Any reason why seed ripening would slow down with increasing temperature? I am missing any justification. If fruiting would be advanced, then I would expect seed dispersal also to be advanced. This is actually also what the data shows (no surprise).

RESPONSE: Thanks for noting this point of confusion. Upon further consideration, we agree that the fruiting scenario put forth in Fig 1 is unlikely and have made several changes to update. For context, our intention with this scenario was to highlight differences between spring (‘early’) and autumnal (‘late’) phenophase responses. We proposed that ‘early season’ phenophases would advance and ‘late season’ phenophases would be delayed, as most studies suggest an advancement in spring phenology and either no change or a delay in autumn phenology (Piao 2019). We also wanted to propose that “Asynchronous shifts in early and late season phenophases may result in the lengthening or contracting of the growing, flowering and/or fruiting seasons” (L89-91).

However, because fruiting is neither an early nor late season phenophase, it does not fit clearly into this scenario, and thus we have moved fruiting to the middle of the figure. We follow the same line of reasoning for the end of flowering, and thus these phases no longer have a prediction on the direction of their shifts for this scenario. Nonetheless, a shift in either one or both of the ‘paired’ phenophases results in a lengthening of the growth, flowering and fruiting periods. We have added the terms “spring”, “autumn”, respectively into lines 85-86 in the introduction and in the figure legend to clarify.

Most of the research on these questions of early vs late season phenology have been for leaf and flowering and very little for fruiting phenology. There is some evidence that the duration of the fruiting period is either static or increases slightly with warming, but that the increased duration is driven mostly by a delay in the end of fruiting (Jiang et al 2016). In the Jiang study, 2 out of 6 species examined had significant changes in both the timing of first fruiting and last fruiting in response to warming, where first fruiting shifted slightly earlier and last fruiting shifted later (Fig 3, Appendix S2). The suggested explanation for this response is not that seed ripening ‘slows down’ *per se* with increasing temperature, but rather that seed size may grow larger with a longer development time which could improve seedling establishment and growth. We have also added the Jiang et al. 2016 reference to the introduction as an example of both asynchronous and fixed responses (ref 39).

As the reviewer mentions these patterns are not what we observe in our fruiting data, however we present these as hypothetical scenarios. Broadly, there are still many unknowns regarding fruiting phenology responses to warming and how they may differ from flowering phenology

responses so we hope that this conceptual framework can help promote further discussion on these topics.

3) Snow. OTCs typically accumulate (a lot of) snow. For those sites where OTCs were deployed year round, this could be an important confounding factor. Snow melt is an important driver of plant phenology in tundra ecosystems. Yes, year-round deployment may lead to increased thermal sums (L138), but snow accumulation could have a much stronger (and potentially opposite) effect. For which of the sites were the OTCs deployed year round, and to what extent did this affect snow melt date? Also, in this context, hypothesis 4 needs to be better introduced.

RESPONSE: We agree that snowmelt timing can be an important confounding factor on the influence of OTC warming. Unfortunately, we have incomplete snow melt data from the different sites and across years. This is why we did not include snow melt date directly into our models, but rather used OTC deployment period to estimate how OTCs may interact with snow melt when influencing phenology, with the prediction that OTCs would create earlier snowmelt than control plots at year-round OTC sites (Danby & Hik 2007, Dabros et al 2010), but not at summer-only sites as they are placed on after snowmelt.

OTCs were deployed year-round at 8 sites and 24 subsites and of these we have snowmelt data from 3 sites, 15 subsites and 29 site-year combinations (n=2246 observations). For these subset of sites, snowmelt date was significantly earlier in OTCs than in control plots by 1.02 +/- 0.17 days (95% CI= 0.68, 1.36 days). This is in line with our prediction that snow melt can be advanced in OTCs deployed year-round due to increased thermal sums and higher solar radiation. We have added this model summary information to the supplement (Appendix S4), and information on snowmelt data to the methods (L381-384) and the available snowmelt data is now posted to the github repository. We have also added an additional table to the supplement which shows the sites with year round versus summer only deployment periods (Table S1b) in addition to snowmelt data availability. Finally, we have added 'and earlier snowmelt' to hypothesis 4 (L142) and a discussion of these results in lines 288-293.

4) As I understand, all phenological responses presented in this manuscript are based on the average of all selected plant species. However, this has not been clearly explained. Also, from the Methods section it is unclear *how* the plant species were selected and *how many* plant species were selected. It is only in the Discussion that I learn that only a subset of the species in the community was sampled. Much more information is needed.

RESPONSE: We have added the following information regarding how species were chosen in the Methods lines 386-392:

"Plant species monitored at each site followed criteria as defined in the ITEX manual (Molau and Molgaard 1996) including prioritizing one or more of the following circumpolar main 'target' species: *Carex aquatilis*, *Cassiope tetragona*, *Dryas integrifolia*, *Dryas octopetala*, *Eriophorum vaginatum*, *Oxyria digyna*, *Bistorta vivipara*, *Ranunculus nivalis*, *Salix arctica*, *Salix herbacea*, *Salix polaris*, *Salix reticulata*, *Saxifraga oppositifolia*, *Silene acaulis*". Fifteen of the 18 sites

included one or more of these circumpolar species, while 3 sites included one or more dominant tundra plant species present at their sites but not on this list (see Appendix S2).

For more context, sites followed this selection criteria to the best of their ability based on plant species composition at different sampling locations. 15 out of 18 sites in our study include at least one species in the priority list, while several sites include multiple. The 3 sites which do not meet this criteria are more recently established alpine sites (Jakobshorn, Gavia Pass, and White Mountains). Jakobshorn includes congeners of species in this list (*Carex* spp.) as well as circumpolar species not in this list but measured at several other sites (e.g. *Empetrum Nigrum* and *Vaccinium Uliginosum*). Gavia Pass and White Mountains simply chose dominant plant species at their sites, and we retained these sites in order to maximize spatial/geographic coverage in our analysis, as species differences were accounted for in our random effects structure. These sites combined represent a very small proportion of the total dataset (~1.4%), but nonetheless species' responses at these sites were consistent with broad patterns (earlier flowering, end of flowering, seed dispersal). Of key importance to this study is that species which were selected at the start of an experiment were consistently monitored at each site over time.

5) The assumption that evergreen and deciduous species do not differ in their leaf phenology does not make any sense. Evergreen and deciduous species per definition differ in leaf phenology. So, these two groups of species should be considered separately.

RESPONSE: Thanks for noting this point of confusion. Our intention was not to signify that deciduous and evergreen species do not differ in their leaf phenology, simply that we did not separate them in our modeling approach. This is because while evergreen species in the Arctic (mostly heath shrubs) do retain some green leaves throughout the winter, they also undergo leaf color change which can be monitored in the same way as deciduous species (see Livensperger et al. 2019). Furthermore, definitions used to measure leaf phenology are consistent for each (sub)site and species (Appendix S2) over time and we include species specific random effects in our models to account for differences among species.

To confirm this approach, we ran leaf green up and senescence models with an interaction term between treatment and leaf habit (deciduous, evergreen). We found no significant difference between evergreen and deciduous species in their timing of leaf phenology overall or in response to OTC warming as the confidence intervals for both leaf habit and leaf habit x treatment crossed zero. We have added these results to the supplement in Table S2 and clarified our reasoning in the methods lines 402-405.

6) More generally, I understand that the dataset is complicated and that the analyses depend on a lot of assumptions and uncertainties (e.g., L356-374 and L377-394), but without taking into account important variation among sites, species, etc. the results are simply not very informative. One of the strengths of the ITEX network is the availability of all sorts of ancillary data. These data should be better utilized.

RESPONSE: We are very interested in taking advantage of the ancillary data in the ITEX network such as information on snowmelt and degrees of OTC warming. Unfortunately, much of

this ancillary data is not collected uniformly across sites and/or at all sites, which greatly limits our ability to include these data in a large synthesis (Also see comments 3, 9 & 10). There is always a tradeoff between detailed, site-specific studies which can address narrower mechanisms (as done previously for some of these sites) and large scale (global/multi-decadal) syntheses to detect broad patterns and responses.

Nonetheless, we have included and estimated with full uncertainties the variation among sites, species, subsites (experimental locations) within sites and years within sites in response to experimental warming through the random effects structure in our hierarchical models. All of these parameter estimates can be found in the Appendix tables S3 a-d. In addition, we take into account and further assess how site level characteristics such as latitude, local climate, and soil moisture may interact with the effects of experimental warming. We find overall that these interactions are minor (Table 3, S2).

Per the reviewer's request, we have also added information and run additional models utilizing ancillary data regarding site level snow melt dates and deciduousness of species (see comments 3 and 5). We feel that these additional analyses have strengthened the paper and further supported this and future research. For snow melt, the confirmation that snowmelt is indeed earlier in year round OTC plots (for the sites we have data for) provided quantitative support to the patterns we predicted in our hypothesis and shown in other studies. For leaf habit, the confirmation that deciduous and evergreen species in the Arctic do not inherently differ in their timing of leaf phenology or their responses to OTC warming, provides valuable information to further studies in tundra phenology.

7) What are 'subsites'? The study sites are nicely indicated on the map in Fig. 2, but nowhere in the manuscript I can find anything about 'subsites'. What are they, how are they selected/analyzed, and why is it this important?

RESPONSE: Subsites are experimental locations replicated within sites as stated in lines 367-368 of the methods. These locations (i.e. where long-term OTC plots are established) are defined by the site Principal Investigators, often chosen based on plant community type (eg tussock tundra, dry meadow), sometimes by hydrology (wet, dry) and sometimes by the dominant plant species (i.e. Cassiope, Dryas). Other times they are chosen at random. Some sites only have one experimental location and thus only one 'subsite.' This determination is left up to the site PIs, but the locations of experimental 'subsites' remain consistent over time. The number of subsites at each site is listed in Table 1 and we have added further information regarding each subsite in Table S1b.

This is important because it highlights the nested structure of the experimental design, and allows us to consider spatial and environmental heterogeneity within and across sites, by incorporating both site and subsite within site random effects in our models. In addition, because soil moisture can vary greatly across the landscape (within sites), we incorporate information regarding soil moisture at the subsite level. Nonetheless, overall variation in phenology responses between sites is much larger than variation between subsites within sites

(Table S3). Model estimates for all subsite within site combinations can be found in Appendix S3c.

8) Why would phenology be less variable in OTCs than in control plots (L173-177)? Would this have anything to do the shape of temperature response curves? Some more explanation here would be helpful.

RESPONSE: Thanks for noting this point of confusion. Indeed this sentence was an incorrect interpretation of the model hyperparameters (sd Intercept and sd treatment-Table S3). Upon further investigation, the comparison of these parameters within a group is itself not informative, only the comparison of a parameter across groups. For example, comparing the variation in site level intercepts (sd Intercept) to the variation in site level slopes (sd treatment) is arbitrary and rather the ('cor') parameter is intended to provide the comparison between random slopes and intercepts within a group (Gelman and Hill, 2007). A positive correlation suggests that groups with larger intercepts also have larger slopes, and vice-versa. None of the correlation estimates were significantly different from zero in our models, however, so we have removed any discussion of comparisons between random slopes and intercepts.

We have corrected this section of the results (L177-181), removing the statement about variation in OTCs versus ambient, and updated the Table S3 legend for clarity.

9) The effect of OTCs on air temperature is highly variable among sites (see Tables S1). So, in explaining the large variation in plant phenological responses to OTC warming, it would make sense to actually use this information. Lumping responses to +2.3C in Finse (Norway) with responses to 0.5C in Daring Lake (Canada) is an over-simplification and masks ecologically-relevant differences among sites. Why would you use 'OTC warming' as a categorical variable (yes, no) if you know that in reality is it a continuous variable?

RESPONSE: We agree that a continuous warming predictor is preferable to a categorical variable (warming vs control), however, as described in lines 374-378 of the methods: "Because sites measured the degrees of warming achieved in OTCs at different time periods we are unable to accurately estimate phenology shifts per degree of warming, however the range of warming achieved is well within the projected climate warming for tundra ecosystems, providing a realistic, though likely conservative, estimate of future scenarios."

For more context, some sites measure the difference in air temperature achieved in OTCs across multiple years and/or across the entire growing season while others measure in only one year or only at the peak of the growing season. Additionally, some sites measure at the subsite level to capture differences across the landscape while others only at the site level. Thus, values in Table S1a are rough estimates of the average level of warming achieved over time at a given site. We did not find these data to be comparable enough across sites and years to include directly in our models. For these reasons, we use a categorical variable of warming/control, and then include other important environmental covariates (site level climate, latitude, soil moisture) to help further explain differences among OTCs across sites.

10) The suggestion that early and late season phenological events may differ in their responses to warming 'if they are co-limited by different non-temperature variables' (L84) is interesting.

However, other than grouping the sites in 'dry', 'moist', and 'wet' (using unclear criteria; see comment nr. 11), this idea remains unexplored. Given the large amount of environmental data collected within the ITEX network, I would expect more in-depth and more direct analyses of how 'non-temperature variables' mediated the OTC effect sizes on plant phenological responses. Here, snow melt would be a good start (see comment nr. 3). This would allow to dig deeper into the mechanisms.

RESPONSE: Thanks for this suggestion. It is true that sites in the ITEX network collect a large amount of environmental data, however (unfortunately) much of these data are not collected uniformly across sites and/or at all sites, which greatly limits our ability to include these data in a large synthesis (also see previous comment). As our objective for this study was to understand the impacts of experimental warming on tundra-wide phenology, we included as many sites with experimental OTCs and phenology data as possible, without constraining based on availability of other environmental measurements.

We have included as much ancillary information as we could to help capture differences in non-temperature variables and test their interactions with OTC warming. These include: 1) soil moisture, as the reviewer mentions-see comment 11 for clarification 2) latitude which is a proxy for day length/photoperiod and other climatic variables (eg precipitation type, permafrost etc.), 3) OTC deployment period which can influence snow melt timing 4) Length of experimental warming which provides information on temporal accumulation and lag effects and 5) site level ambient climate-still temperature, but at much finer spatial and temporal scales.

Based on the reviewer's suggestion we have now added discussion and additional data on snowmelt to the manuscript (see response comment to 3) although this dataset is incomplete for similar reasons described above. Ultimately, and to our surprise, very few of the environmental covariates we tested had significant interactions with OTC warming (only 3 out of 42 interactions tested), and for this reason, we focus less on the influence of 'non-temperature' variables in the Discussion.

11) 'Soil moisture' was used as one of the explanatory variables (e.g., Fig. 4). But, how was 'soil moisture' measured? And how were sites categorized? This is not a trivial task. Nothing is mentioned about this.

RESPONSE: Soil moisture categories were defined by site PIs based on an average gravimetric water content (GWC) at each site <20% (dry), 20-60% (moist), >60%(wet) GWC. We have added this information to the methods in lines 378-380 and added information on soil moisture content for each subsite to Table S1b.

12) For the tested 'Year' effects it is unclear what temporal scale this applies to. While the Abstract says that data for 20+ years were used, nowhere in the manuscript I can find any information on this. For all species, were phenological data collected every year for 20+ years? Probably not. Discussion on local adaptation, acclimation, etc. does not make sense if the temporal scale and resolution is unknown. More details and justification is needed.

RESPONSE: We included the effect of 'year' in two ways:

1. The specific calendar year in which phenology measurements were taken at a given site (i.e. year within site random slopes-Appendix S3d).
2. The number (i.e. sum) of years of experimental warming that had occurred at a given site at the time of each phenology observation (i.e. Years of warming-Table S2).

The first is estimated as a random slope of year nested within site as described in the methods (L480-481) and in Appendix S1 and all model estimates can be found in Appendix S3d. The second is the covariate of years of warming (i.e. warming duration), which was tested as an interaction with treatment. To clarify, we have added "calculated as number of years from the start of the experiment at each site/subsite" to the description of years of warming in lines 496-497 in the methods.

The largest number of years contained at a single site was 20 (Table 1), but within phenophases, we have 27-28 years of data across multiple sites (Table 2). This is why we had used the terminology '20+ years' previously, but we agree that it was confusing. Thus we have changed this terminology throughout the manuscript to say "with observations from the last three decades."

3) The conclusions about potential consequences for trophic interactions, C cycling, and ecosystem functioning are rather suggestive and are not based on the data presented in this manuscript. The 'significant consequences' of phenological shifts by a day or two are not well substantiated. Paragraph L252-262 is full of uncertainties, and in the end it seems that nothing can be concluded regarding the impacts on global C stocks. Also, the manuscript does not contain any C data. This is despite the large amount of C data collected within the ITEX experimental network.

RESPONSE: We understand the reviewer's concerns and have toned down/clarified the language regarding significant trophic interactions and ecosystem consequences in several parts of the Discussion. First, we removed the statement "with potentially significant consequences for trophic interactions and ecosystem function" from the first paragraph of the discussion and replaced it with the following (L197-202):

"Deciphering the effects of warming temperatures on plant phenology using only observational data can be challenging due to inherent correlations in environmental drivers (Hanninen et al 2019) yet very few studies have assessed phenology responses to experimental warming at this spatial and temporal scale (but see Ettinger et al. 2021). In addition, our estimates are likely very conservative, as actual levels of climate warming in the Arctic are predicted to reach 3-5x the magnitude of warming achieved in OTCs (Overland et al 2013)."

We have also clarified and added clearer examples to the paragraph describing potential impacts on trophic interactions and in lines (223-226):

"Despite the relatively small magnitude shifts observed in our study, plant and pollinator interaction networks can shift significantly on very short term (day to day) scales (Caradonna et

al. 2020, Frund et al 2011) and influence the stability and feasibility of ecological communities particularly for ecosystems with very short growing seasons such as the Arctic Tundra (Saavedra et al. 2016, Song and Saavedra 2018)”

We also removed the sentence about tri-trophic interactions from this paragraph.

In terms of carbon, we believe that quantifying the effects of phenology shifts on C stocks at the plot level would be an exciting next step in this research, and we have proposed this in the section on future ideas for this research (L359-361)

As of now, we do not have the data to quantify the impact of these phenology shifts on global C stocks. Rather, we aim to place our results in the context of other modeling studies which have estimated links between shifts in growing season length of a similar magnitude and GPP across the northern hemisphere (Piao et al 2007, White et al 1999). Growing season length is arguably the strongest predictor of primary productivity for the tundra biome (Ueyama et al 2013), and thus we feel confident that our robust estimates of shifts in green up and senescence leading to a projected 1.5 day increase in growing season length (Fig S1), will have impacts on C stocks in the tundra, but we are not attempting to quantify what magnitude those impacts will be.

We have clarified this by removing any numeric estimates from the paragraph and using clearer language when discussing results of other papers versus our own. We also removed the last sentence “Furthermore, these estimates are likely very conservative, as actual levels of climate warming are predicted to be stronger than the 0.5-2.3 degrees of warming achieved in OTCs across sites (Table S1).” as we realized this was confusing, as it was not referring to C estimates but rather to growing season length estimates.

14) In the Conclusions section, the extrapolation based on *linear* temperature responses makes little sense. For many reasons, temperature responses will certainly not be linear.

RESPONSE: We agree that a linear extrapolation may be problematic, and we have removed this part of the conclusions. We have replaced it with a clearer estimate given the data we do have, stating in lines (328-333):

“In addition, divergent shifts in leaf green up and senescence led to a 1.5 day increase in species’ growing season lengths, or approximately 3% of the average 50 day growing season length, given this modest level of warming. We consider these estimates to be on the low end of potential shifts in phenology, as much higher levels of warming (~3-13 C) are expected in the Arctic by the end of the century.”

15) In the Conclusions section, rather than just suggesting that deciphering the potential drivers and ecosystem level consequences is important (we’ve known that for many years), it would be much more interesting to propose a way forward on *how* to do that.

RESPONSE: Thanks for the suggestion. We have updated this section (L351-361). It now reads:

We suggest several ways to improve future research and decipher the potential drivers and ecosystem level consequences of phenology responses to warming across the tundra: 1) Directly testing the physiological cues of vegetative versus reproductive phenology through growth chamber or field manipulations (warming leaves/flowers only) 2) Co-measuring individual plant phenology alongside timing of pollinators and assessing outcomes on plant reproductive fitness (seed production and viability) 3) Monitoring phenology at the plant community level, where all members (not just dominant or charismatic species) are recorded. 4) Using a gradient of experimental warming treatments (extreme, moderate, mild) to understand the consequences of more severe warming and limits of linearity of responses to temperature and 5) Measuring the relationship between changes in phenology and ecosystem C fluxes using *in-situ* plot level measurements and biomass estimates and/or site level Eddy covariance.

Reviewer #2 (Remarks to the Author):

The authors of this report analyse a large dataset including very large spatial and temporal scales to predict ecosystem responses to warming using open-top chambers (OTCs) across Arctic, sub-Arctic, and alpine ecosystems gathered by the ITEX consortium in 1992-2019.

They show that OTCs influence plant phenology, including phenophase shifts in the range of 0.7-2.7 days in response to the 0.5-2.3°C of warming achieved with chambers. However, the high variability of environments prevented to accurately estimate phenology shifts, as OTCs and sites were quite diverse. In addition, phenology definitions slightly differed across subsites and species, and census intervals also varied, introducing more noise in the data. Handling such heterogeneous dataset is commendable, and despite these difficulties, the authors got the most out of the available data.

RESPONSE: We thank the reviewer for their careful reading of the manuscript and thorough comments. We have addressed each of the reviewer's comments below (in red) and feel the manuscript is thus improved. All updates to the manuscript and supplement are highlighted in yellow.

However, including all sites in the same analyses may only blur the outcome. I wonder whether the different sites or subsites could be grouped to get more meaningful results. I am sceptical that climate in the Alps or Sierra Nevada in California is comparable to the climate in Svalbard or Alexandra Fiord. In addition, by including a large number of plant species assuming that all behave the same way might be an important confounding factor, as they respond in quite idiosyncratic ways to environmental clues.

RESPONSE: We agree that site level differences are an important consideration. Rather than categorically group sites into 'biomes', for example High arctic, Low arctic and Alpine, which has been done in other studies, we used a continuous predictor of latitude and site level ambient climate as co-factors into our analyses. This was because for many sites, biome type is somewhat unclear, for example alpine sites in Norway or Sweden such as Latnjajauare and

Finse are likely to be more similar to a low arctic site than an alpine site. And some low arctic sites such as Daring Lake, Canada had higher average temperatures than even alpine sites (Fig S2). So we used quantitative variables of latitude and site level ambient climate as predictors. To our surprise we found no significant interaction between latitude and experimental warming and we only found an interaction between site level ambient climate and experimental warming for seed dispersal (Fig S2).

In addition, the reviewer's concern highlights the strength of the Hierarchical Bayesian approach we used in that in the same model we estimate **both** the overall responses to experimental warming (lumping all species and sites, subsites, Table 3) **and** the response of each species, site, subsite and site-year to experimental warming (Appendix S3 a-d). While sites and species definitely differ in the *magnitude* (and thus significance) of their responses, there are no site or species estimates which differ in the *direction* of their response from the overall patterns, giving us confidence in our results and conclusions.

I am not familiar with some statistical procedures, particularly Bayesian methods, but analyses seem to be sound. The reported rates of change in phenophases differ between control and OTC treatments, although these differences are small, and much larger than inter-annual variations. Thus, I wonder whether such small changes may be meaningful.

RESPONSE: Thanks for this comment. We agree that the rates of change are somewhat small, but we consider these to be conservative estimates given that the degree of warming achieved in OTCs (~1-3 degrees C) is lower than the projected levels of climate warming in the Arctic. We have added text to the discussion highlighting this line of reasoning (L200-202) and (L331-333).

Nonetheless, given this modest amount of warming, we do observe shifts that we consider substantial. Previous work suggests that even changes at this small scale (~1 day) can be meaningful for ecosystem processes such as C cycling and GPP across the Tundra (Piao et al 2007) and plant-pollinator interaction networks (Saavedra et al 2016), as we discuss in lines 268-270 and 223-226 respectively.

We have added citations providing examples of how shifts at this scale can influence plant outcomes (Saavedra et al 2016 and Song & Saavedra 2018) showing how stability in plant-pollinator interaction networks can be influenced at a day-to-day scale, with empirical evidence from an Arctic tundra site in Greenland. In addition, we have modified our language when discussing implications for global C stocks and added a relevant citation showing the strong relationship between growing season length and GPP across the tundra (Ueyama et al 2013). Thus we feel that our robust estimates of shifts in green up and senescence leading to an estimated 1.5 day increase in growing season length will have impacts on C stocks in the Tundra, but we are not attempting to quantify what magnitude those impacts will be.

We have also added information to estimate the average number of days between phenophases based on model intercepts in Table S4 to place the magnitude of these changes into context given their respective phenoperiods (L526-529). We now highlight these comparisons in the discussion in several places and in the abstract (L 20):

“Our estimates predict an increase in species’ growing season lengths by approximately 1.5 days (Fig 3, Fig S1), which reflects a 2.5-3.75% increase in the 40-60 (50 ± 10) day average length between leaf green up and senescence across all sites in this study (Table S4) (L264-266).”

“In addition, divergent shifts in leaf green up and senescence led to a 1.5 day increase in species’ growing season lengths, or approximately 3% of the average 50 day growing season length, given this modest level of warming. (L 328-331).”

“Thus a 2-2.4 day shift in flowering time may be especially relevant for tundra species with rapid flower development, as the average amount of time between start and end of flowering across all sites was only 18 (± 8) days (L231-234).”

I think the manuscript is well written. The results are properly discussed in the context of previous literature, but I think there is some speculation regarding response mechanisms, as they were not addressed in the experiments and require physiologically-oriented approaches. Therefore, this part of the discussion should be drafted more carefully as there are no data to support claims.

RESPONSE: We have more carefully worded this section of the discussion in lines (209-211) and clarified that understanding the mechanisms driving these responses requires specific testing that is beyond the scope of this study. It now reads:

“However, deciphering the mechanisms driving these differences is difficult and requires physiological approaches which are beyond the scope of this study. Previous work addressing these questions has generated mixed results.”

We have also added references to how Hypotheses 1 and 2 correspond with the different scenarios in Fig 1 and removed the potential mechanisms driving these hypothesized shifts, as we have not directly tested them in this study.

Overall, the net contribution of this paper seems not so novel to me, since similar findings have been reported in the last 10-15 years from a diversity of environments, mostly temperate but also high-latitude environments. None of them had, to my knowledge, the strong experimental support used in this paper. Although the reported results support earlier claims concerning phenology responses to climate change, they would certainly be of interest to others in the field. Even if largely confirmative, the data reported in this paper will consolidate knowledge in the field.

RESPONSE: We agree that much of the novelty of this paper comes from its consolidation of knowledge at the tundra scale. We also feel this study is particularly novel and timely for the following reasons:

1. This study focuses on tundra plant phenology and to our knowledge is the largest scale analysis of tundra phenology to date. Tundra phenology has been largely understudied in terms of phenology research, second only to the Desert biome (Diepstraten et al.

2018) and thus calls for larger synthesis of this important biome have recently been made. We feel this work addresses this important research gap, in particular the need for multi-site syntheses.

2. Our study focuses on plant phenology responses to experimental manipulations. The need for experimental (not simply observational) estimates of phenology shifts in order to accurately parameterize process based-phenology and global change models has been highlighted in recent reviews (Hanninen et al 2019). This allows us to generate an understanding of the influence of warming temperatures independent from other environmental drivers.
 - a. This is particularly critical for late season phenophases such as leaf senescence and fruit maturation/seed dispersal, for which experimental data are greatly lacking. Our study includes the most robust experimental estimates of late season phenophase responses to warming in the Tundra biome to date.
3. This study incorporates all major plant phenophases across the growing season, including the beginning and end of leaf, flowering, and fruiting phenology. To our knowledge, this is more than any other synthesis we are aware of, as many multi-site syntheses analyze one or two plant phenophases (often leaf green up and/or flowering). For example, Prevey et al (2019) focused on flowering and end of flowering only.

We have more thoroughly incorporated these key points of novelty and how our study moves beyond previously published work throughout the manuscript in the abstract (L9-11) , introduction (L123-125) and discussion/conclusion (L198-200, 323-326, 335-337) .

Reviewer #3 (Remarks to the Author):

By using an impressive dataset derived from the ITEX initiative, this study analyze detailed both reproductive and vegetative phenological records taken on tundra plant species exposed to experimental warming in 18 sites and 46 sub-site locations. On each location experimental warming was generated passively by using open-top transparent plexiglass chambers, containing vegetation patches formed by different tundra species. The duration of the warming treatment differed among sites, while in some sites phenological records has been taken for 27 yrs. in some sites there were only one or 2 yrs. of records. The main question addressed with these data set were: 1. What is the overall magnitude (number of days) and direction (advance or delay) of plant phenology shifts in response to warming? 2. Does warming differentially affect reproductive and vegetative phenology? 3. Does warming shorten, lengthen, or have no effect on the duration of growth, flowering and fruiting/seed maturation periods? 4. How do plant responses to warming vary across spatial and temporal gradients in resource availability and ambient climate? 5. Are plant responses to warming sustained over time?

In general, this is very interesting study, addressing important questions of general interest in the scientific community. Nevertheless, although I am very sympathetic with this study, I think that there are some important points that needs to be addressed.

RESPONSE: We thank the reviewer for their careful reading of the manuscript and thorough comments. We have addressed each of the reviewer's comment below (in red) and feel the

manuscript is thus improved. All updates to the manuscript and supplement are highlighted in yellow.

First, it is not totally clear to me the step forwards in our knowledge of the tundra plant phenological responses to warming considering the previous contributions of Prevey et al. (2019: Warming shortens flowering seasons of tundra plant communities. *Nature Ecology & Evolution*, 3(1), 45-52.) and Myers-Smith et al. (2019: Eighteen years of ecological monitoring reveals multiple lines of evidence for tundra vegetation change. *Ecological Monographs*, 89(2), e01351). It is important that authors can clearly state what is really novel in this study.

RESPONSE: We believe this study consolidates knowledge in the field and is thus novel and timely for several reasons:

1. This study focuses on Tundra plant phenology and to our knowledge is one of the largest scale analyses of tundra phenology to date. Tundra phenology has been largely understudied in terms of phenology research, second only to the Desert biome, (Diepstraten et al. 2018) and thus calls for larger synthesis of this important biome have been made. We feel this work addresses this important research gap, in particular the need for multi-site syntheses. For example, Meyers-Smith et al 2019, used an impressive dataset as well, but only from one site, Qikitaruk (Greenland) whereas our study includes 18 sites. (Also, our study includes experimental warming manipulations where Myers-Smith does not-see next point).
2. Our study focuses on plant phenology responses to experimental manipulations. The need for experimental (not simply observational) estimates of phenology shifts in order to accurately parameterize process based-phenology and global change models has been highlighted in recent reviews (Hanninen et al 2019).
 - a. This is particularly critical for late season phenophases such as leaf senescence and fruit maturation/seed dispersal, for which experimental data are greatly lacking. Our study includes the most robust experimental estimates of late season phenophase responses to warming to date in the Tundra biome.
3. This study incorporates 6 plant phenophases, including both reproductive and vegetative, across the growing season. To our knowledge, this is more than any other large synthesis we are aware of as many multi-site syntheses analyze one or two plant phenophases (often leaf green up and/or flowering). For example, the synthesis by Prevey et al 2019 used a large global dataset (similar to ours), but only included flowering and end of flowering phenophases. Thus, this study makes a significant contribution beyond Prevey et al 2019.

We have more thoroughly incorporated these key points of novelty and how our study moves beyond previously published work throughout the manuscript in the abstract (L9-11) , introduction (L123-125) and discussion/conclusion (L198-200, 323-326, 335-337) .

Second, according to the information provided on Table 3, none of the sites contain 27 yrs. of records for the different phenophases; only 4 of them have a data set longer than 10 yrs., and 9

longer than 5 yrs.; indeed, 5 sites have only 2 or less yrs. of records. Thus, I'm wondering how this asymmetry in the duration of the phenological records can affect the results on either direction. For instance, what if short-term records buffer the long-term responses generating less dramatic changes; or the other way around: short-term records leveraging the observed changes? I think that is important that authors make clear how do they managed the asymmetry in the duration of the phenological records among locations. This is also important for the assessments related to question 4.

RESPONSE: We agree with this concern and because of this we directly tested whether the length of the warming experiment (i.e. Years of warming) interacted with the effects on phenology shifts. Surprisingly, we found no significant interactions (Table S2), suggesting to us that responses to warming experiments are not different between short and long time intervals. We discuss this further in lines 312-320.

To clarify, the largest number of years contained at a single site was 20, however within phenophases, we have 27-28 years of data (see Appendix S3d for site years). This is why we had previously used the convention "20+ years". We have changed this terminology throughout the manuscript to say "with observations from the last three decades".

Third, the observed changes in some phenophases were within 1-3 days, which are regarded as considerable by the authors. However, I miss more a thorough discussion considering strong examples that shifts of 1-3 days have important consequences for plants. Arguments like those provided in lines 208-2016 are speculative and do not provide unequivocal examples of how shifts of 1-3 days really affect plant performance.

RESPONSE: We agree that our estimates of 1-3 day average shifts in phenology are modest, and we have changed the word considerable to 'measurable' in line 184.

We have also made several updates to try to better contextualize these shifts. First, we have added information on model intercepts to show the average dates (DOY) that each phenophase occurred across all sites, species, and years (Table S4). With this information, we estimate the average growing season length to be 50 +/- 10 days and a flowering season length of 18 +/- 8 days. Thus we estimate a lengthening of the growing season to represent a shift of 2.5-3.75% of the total and a 2-2.4 day shift in flowering to also be relevant given such a short flowering period. We highlight these comparisons in several places in the discussion (L264-266, L231-234, L328-331) and in the abstract (L 20):

Next, we have added clearer examples of how shifts as this scale can influence plant outcomes including work by (Saavedra et al 2016 and Song & Saavedra 2018) showing how stability in plant-pollinator interaction networks can be influenced at a day-to-day scale, with empirical evidence from an arctic tundra site in Greenland . In addition, we have modified our language when discussing implications for global C stocks and added relevant work showing the strong relationship between growing season length and GPP across the tundra. Thus we feel that our robust estimates of shifts in green up and senescence leading to an estimated 1.5 day increase

in growing season length will have impacts on C stocks in the Tundra, but we are not attempting to quantify what magnitude those impacts will be.

Finally, while they are somewhat small, we consider our 1-3 day estimates to be highly conservative, the degree of warming achieved in OTCs (~0.5-2.5 degrees C) is much lower than the projected levels (and in some cases even current) of climate warming in the Arctic. We have added text to the discussion highlighting this line of reasoning (L200-202) and (L331-333).

Reviewers' Comments:

Reviewer #1:

Remarks to the Author:

15 March 2021

Experimental warming differentially affects vegetative and reproductive phenology of tundra plants

I have reviewed this manuscript before, so I will take it from there.

I was a bit critical of the original submission, but the authors did a great job in revising their manuscript and addressing my and the other reviewer's concerns. Much appreciated!

The manuscript has much improved and it reads fine now, and I do not have much to comment on.

The novelty of the study has been made clearer and, overall, the strengths and weaknesses of the dataset have been better explained.

I am happy to see corrections made to Figure 1; the end of flower and fruiting are indeed better considered as mid-season phenophases. I am also happy to see some new analyses of snowmelt data. And, it is good to see a description of the plant species selection.

I understand that using more ancillary data is difficult because most data have not been uniformly collected across sites. This is unfortunate and it seems a major shortcoming of the otherwise great ITEX network. What is the point of setting up a network of sites if there is no coordination among collaborators on what data they collect? This particularly applies to baseline data such as the effects of the OTCs on soil and air temperature.

I still think that mentioning 'observations over the last three decades' in the Abstract and elsewhere in the manuscript is misleading. It reads as if 30 years of data has been used, while in most cases it is only 1 to 6 years. It is an really nice data set on plant phenology, but please do not make it sound better than it is.

L353 though = through

Reviewer #2:

Remarks to the Author:

I think the authors addressed my comments in a satisfactory way and I have no further concerns.

Reviewer #3:

Remarks to the Author:

I have read the revised version of this study as well as the rebuttal letter. In my opinion, authors have done an excellent job in considering all my previous concerns as well as those from the other reviewers. I think that this is highly relevant study that certainly add to our knowledge on the tundra plant responses to climate change.

REVIEWERS' COMMENTS

Reviewer #1 (Remarks to the Author):

15 March 2021

Experimental warming differentially affects vegetative and reproductive phenology of tundra plants

I have reviewed this manuscript before, so I will take it from there.

I was a bit critical of the original submission, but the authors did a great job in revising their manuscript and addressing my and the other reviewer's concerns. Much appreciated!

The manuscript has much improved and it reads fine now, and I do not have much to comment on.

The novelty of the study has been made clearer and, overall, the strengths and weaknesses of the dataset have been better explained.

I am happy to see corrections made to Figure 1; the end of flower and fruiting are indeed better considered as mid-season phenophases. I am also happy to see some new analyses of snowmelt data. And, it is good to see a description of the plant species selection.

RESPONSE: We thank the reviewer for their careful reading of our revised manuscript and are pleased that we have adequately addressed the reviewer's concerns.

I understand that using more ancillary data is difficult because most data have not been uniformly collected across sites. This is unfortunate and it seems a major shortcoming of the otherwise great ITEX network. What is the point of setting up a network of sites if there is no coordination among collaborators on what data they collect? This particularly applies to baseline data such as the effects of the OTCs on soil and air temperature.

RESPONSE: We agree that this is a major shortcoming of the network, and currently protocols are being developed and implemented to help address the lack of standardized metadata across sites. In particular, the protocol described in Parker et al. (2021) for measuring soil carbon and other related soil parameters is being implemented at many ITEX sites beginning this summer.

Parker, T.C., Thurston, A.M., Raundrup, K. *et al.* Shrub expansion in the Arctic may induce large-scale carbon losses due to changes in plant-soil interactions. *Plant Soil* (2021). <https://doi.org/10.1007/s11104-021-04919-8>. S1: Template for Paired Soil and Vegetation surveys

I still think that mentioning 'observations over the last three decades' in the Abstract and elsewhere in the manuscript is misleading. It reads as if 30 years of data has been used, while in most cases it is only 1 to 6 years. It is a really nice data set on plant phenology, but please do not make it sound better than it is.

RESPONSE: We have changed this to say 'between 1-20 years in duration' for clarity.

L353 though = through

RESPONSE: We have updated this

Reviewer #2 (Remarks to the Author):

I think the authors addressed my comments in a satisfactory way and I have no further concerns.

RESPONSE: We thank the reviewer for their careful reading of our revised manuscript and are pleased that we have adequately addressed the reviewer's concerns.

Reviewer #3 (Remarks to the Author):

I have read the revised version of this study as well as the rebuttal letter. In my opinion, authors have done an excellent job in considering all my previous concerns as well as those from the other reviewers. I think that this is highly relevant study that certainly add to our knowledge on the tundra plant responses to climate change.

RESPONSE: We thank the reviewer for their careful reading of our revised manuscript and are pleased that we have adequately addressed the reviewer's concerns.